# Systematic Multiscale Models to Predict the Compressive Strength of Cement Paste as a Function of Microsilica and Nanosilica Contents, Water/Cement Ratio, and Curing Ages

Chiya Y. Rahimzadeh [1,2,*], Ahmed Salih [3] and Azeez A. Barzinjy [4,5]

1 Civil Engineering Department, Faculty of Engineering, Soran University, Soran 44008, Iraq
2 Scientific Research Centre, Soran University, Soran 44008, Iraq
3 Civil Engineering Department, College of Engineering, University of Sulaimani, Sulaymaniyah 46001, Iraq; ahmed.mohammed@univsul.edu.iq
4 Department of Physics, College of Education, Salahaddin University-Erbil, Erbil 44001, Iraq; azeez.azeez@su.edu.krd
5 Department of Physics Education, Faculty of Education, Tishk International University, Erbil 44001, Iraq
* Correspondence: chiya.yousef@soran.edu.iq

**Abstract:** Sustainable construction requires high-strength cement materials that additives with silica content could provide the requirements as well. In this study, the effect of the micro and nano-size of silica on the compressive strength of cement paste using different mathematical approaches is investigated. This study compares the strength of preferentially replaced cement pastes with microsilica (MS) and nanosilica (NS) incorporation by proposing several mathematical models. In this study, 205 data were extracted from the literature and analyzed. The modeling processes considered the most significant variables as input variables that influence the compression strength, such as curing time, which ranged between 3 and 90 days, the water-cement ratio, which varied between 0.4 and 0.85, and NS ranged between 0 and 15%. MS ranged between 0 and 40% based on the weight of cement. In this process, the compressive strength of cement paste modified with NS and MS was modeled using four different models, including the Linear Regression Model (LR), Nonlinear Model (NLR), Multi-Logistic Regression Model (MLR), and artificial neural network (ANN). The efficiency of the suggested models was evaluated using different statistical assessments, such as the Root Mean Squared Error (RMES), the Mean Absolute Error (MAE), Scatter Index (SI), Objective value (OBJ), and coefficient of determination ($R^2$). The findings revealed that the ANN model conducted better performance for predicting compressive strength for cement paste than the other models based on the statistical assessment. In addition, based on the statistical assessment of the sensitivity of parameters, NS had more of an effect on the compressive strength of cement paste, with 6.3% more than MS.

**Keywords:** microsilica (MS); nanosilica (NS); compressive strength; modeling; sensitivity





## 1. Introduction

Cement-based products are composition materials prepared from ordinary Portland cement (OPC), which is of prime significance in all building fields due to its vast usage and functions. A growing world population and relative requirements for infrastructure on one side, and enormous technical and product developments on the other have raised cement demand. In addition to being highly energy inefficient, the processing of cement and fine aggregates also poses a danger to the environment due to the release of unhealthy contaminants, including microparticles and $CO_2$ gases [1]. It is estimated that cement clinkers' manufacturing is part of the substantial basis of the global $CO_2$ gas footprint. The building industry is listed as one of the major causes of $CO_2$ gas emissions. As demand for cement continues to increase, emissions are projected to become more substantial [2]. In addition, the building material strength, reliability, and maintenance costs are other critical

topics of primary concern. Environmental sustainability includes advanced innovations to monitor and minimize $CO_2$ emissions and energy usage and replace cement with supplementary cementitious materials, namely Pozzolans [1]. Diatomaceous earth [2], silica fume [3], volcanic ash [4], bottom ash, and husk/fly ash [5–7] are used in traditional pozzolans. Furthermore, several nanoparticles (NPs), for instance, carbon nanotubes [8], nano metakaolin [7,9–11], nanosilica (NS) [7,11–16], nano-$Al_2O_3$ [17,18], nano-$TiO_2$ [19,20], nano-CuO [21], and nano-$Fe_2O_3$ [22,23], have also recently been investigated with rapid improvements in nanomaterial technology. In cement paste, these materials are mixed to substitute cement ideally to gain the necessary characteristics of the much more inexpensive materials, besides all the higher performance of the cement materials. Some NPs have high pozzolanic characteristics due to a finer particle volume and function as fillers, leading to uniform, dense, and compressed micro-structural materials [16].

Cement compositions' physical, chemical, and mechanical characteristics have been dramatically improved [24]. The ultra-thin NPs penetrate the voids found in the cement mix, leading to the void size being refined, the void volume decreasing, and the mechanism of the capillary in the voids being disconnected. As a result, the water permeability potential of the cement mix decreases, besides a coincident rise in the chemical resistance of the cement structures. The NPs build a micro-compact structure in cement paste that leads to a remarkably dense cement paste and increases the mechanical characteristics of the cement paste [25]. Silica fume (microsilica) has been used in concrete for the manufacture of high-performance cement-based materials. Micro/nanosilica was found to accelerate the hydration process at early ages because of its considerably fine particle size [6]. Positive values are recorded in the literature for systems containing NPs in the existence of MS. Microsilica can display both cement and pozzolanic activity and is very useful in changing the cement paste's early age and hardened age behavior [26]. Through the creation of a micro-level structure supplying for the closure of cracks, the nucleation effect of NPs promotes the hydration response involving the formation of CSH gel. Due to the massive reaction of NPs of silica resulting in early hardening, characteristics such as the setting time are also reduced [27]. In addition to the application methods, the water to cement ratio, the fresh and hardened characteristics depend on the concentration, composition, and particle dimension of the binder and additive particles. It is possible to use NS in powder form or colloidally [28]. Colloidal NS contains amorphous hydroxylated silica NPs with a dimension varying from 1 to 500 nm in a colloidal suspension form. Colloidal NS particles illustrate less segregation and a nicer dispersity in cement paste and are regarded to promote the development of high-rigidity CSH gel [29]. While studies are available on the single use of MS and NS or combined to alter the behavior of cement paste, there is not much evidence on which one of those parameters is most effective on mechanical properties, such as the compressive strength.

The purpose of this study is to find out the effect of the various ratios of amorphous NS and MS on the compressive strength of cement paste as a function of the water–cement ratio and curing time. In this manner, some mathematical prediction models, including the Linear Relational Model (LR), Nonlinear Model (NLR), Multi-Logistic Model (MLR), and artificial neural network (ANN) are used in two situations of training and testing to evaluate the effect of NS and MS on the compressive strength of cement paste.

Although compressive strength relies on several parameters, it is also sensitive to the mixed proportion, so using more sophisticated model techniques in combination with simplified tools and numerical equations is beneficial for engineers and helps reduce the number of laboratory experiments required. Multivariable models may be regarded as an excellent solution in this respect. The primary advantage of these methods is that they enable the development of alternatives and solutions for linear or nonlinear issues in which mathematical models cannot easily explain the connection between the components [30]. Several models for predicting cement-based materials and multi-linear regression analysis models are widely employed [31,32]. This research aims to develop, characterize, and recommend a multiscale model for predicting the compressive strength of cement paste

improved with NS and MS. In this respect, the extracted dataset was extensively tested using different modelization approaches to (i) conduct a comparative study and investigate the impact of varying mix formulations, such as NS and MS, as well as the w/c and curing time, on the compressive strength of cement paste adjusted with NS and MS; (ii) using statistical evaluation parameters, test and find the most reliable model to estimate the compressive strength of cement paste adjusted with NS and MS across all models which are linear, nonlinear, multi-logistic, and artificial neural network relation models.

## 2. Materials and Methods

### 2.1. Methods

In total, 205 experimental data from previous publications [33–37] were gathered. The mortar cubes in those publications were determined according to IS 2250-1981, and their size in four papers [33,34,36,37] was $50 \times 50 \times 50$ mm$^3$, and in just one of them [35], the model size of $(40 \times 40 \times 40)$ mm$^3$ was used. The criteria for collecting the data was based on compressive strength test, experimental data, and updated credible journals from 2017 to 2020, with studies including cement mortar adjusted by MS and NS without other admixtures and subjected to the same curing conditions. The gathered data were evaluated statistically and randomly assigned to one of two groups. The first group, which was larger, included 150 datasets that were utilized to build the models. The second group included 55 datasets used for testing the models. The summary of the cement paste mixes database is reported in Table 1. The present objective of the study was to predict which size of silica additives was more effective, the nano-size or micro-size. They were also reported and analyzed statistically. As can be seen from Table 1, the input dataset included nanosilica content (NS, % of mass of cement), microsilica content (MS, % of mass of cement), w/c ratio, and curing time (t, days). The provided dataset, which included many independent characteristics, was utilized to estimate the compressive strength of cement paste modified with NS and MS. Various models were implemented and compared to the actual measured compressive strength (MPa). Figure 1 depicts the flow diagram for the method used in this research. Additionally, the next parts provided and explained the specifics, including data gathering, analysis, modeling, and testing.

**Table 1.** Summary of different paste mixes modified with nanosilica and microsilica.

| No. | Ref. | w/c Ratio per Mass | Curing Time (Days) | Nanosilica (30–100 nm) (NS) | Microsilica (0.2 μm) (MS) | Compressive Strength (MPa) |
|-----|------|-----|-----|-----|-----|-----|
| | | | | Additives (%) | | |
| 1 | | 0.4 | 3 | 0 | 0 | 15.51 |
| 2 | | 0.4 | 3 | 1.4 | 0 | 16.85 |
| 3 | | 0.4 | 3 | 4.2 | 0 | 23.60 |
| 4 | | 0.4 | 3 | 2.8 | 0 | 21.57 |
| 5 | | 0.4 | 3 | 0 | 4 | 18.88 |
| 6 | | 0.4 | 3 | 2.8 | 4 | 26.97 |
| 7 | | 0.4 | 3 | 4.2 | 4 | 25.62 |
| 8 | | 0.4 | 3 | 1.4 | 4 | 28.31 |
| 9 | | 0.4 | 3 | 4.2 | 9 | 26.29 |
| 10 | | 0.4 | 3 | 0 | 9 | 20.90 |
| 11 | | 0.4 | 3 | 1.4 | 9 | 28.31 |
| 12 | | 0.4 | 3 | 2.8 | 9 | 31.01 |
| 13 | | 0.4 | 3 | 4.2 | 13 | 26.97 |
| 14 | | 0.4 | 3 | 2.8 | 13 | 25.62 |
| 15 | | 0.4 | 3 | 0 | 13 | 23.60 |
| 16 | | 0.4 | 3 | 1.4 | 13 | 24.94 |
| 17 | | 0.4 | 7 | 0 | 0 | 20.90 |
| 18 | | 0.4 | 7 | 1.4 | 0 | 25.62 |
| 19 | | 0.4 | 7 | 4.2 | 0 | 32.36 |
| 20 | | 0.4 | 7 | 2.8 | 0 | 29.66 |

**Table 1.** *Cont.*

| No. | Ref. | w/c Ratio per Mass | Curing Time (Days) | Additives (%) Nanosilica (30–100 nm) (NS) | Microsilica (0.2 μm) (MS) | Compressive Strength (MPa) |
|---|---|---|---|---|---|---|
| 21 | | 0.4 | 7 | 1.4 | 4 | 35.73 |
| 22 | | 0.4 | 7 | 0 | 4 | 23.60 |
| 23 | | 0.4 | 7 | 2.8 | 4 | 34.38 |
| 24 | | 0.4 | 7 | 4.2 | 4 | 33.71 |
| 25 | | 0.4 | 7 | 2.8 | 9 | 39.78 |
| 26 | | 0.4 | 7 | 1.4 | 9 | 35.06 |
| 27 | | 0.4 | 7 | 4.2 | 9 | 33.71 |
| 28 | | 0.4 | 7 | 0 | 9 | 26.97 |
| 29 | | 0.4 | 7 | 0 | 13 | 28.31 |
| 30 | | 0.4 | 7 | 1.4 | 13 | 31.69 |
| 31 | | 0.4 | 7 | 4.2 | 13 | 35.73 |
| 32 | | 0.4 | 7 | 2.8 | 13 | 33.71 |
| 33 | | 0.4 | 14 | 0 | 0 | 22.65 |
| 34 | | 0.4 | 14 | 1.4 | 0 | 29.97 |
| 35 | | 0.4 | 14 | 4.2 | 0 | 37.94 |
| 36 | | 0.4 | 14 | 2.8 | 0 | 35.29 |
| 37 | | 0.4 | 14 | 1.4 | 4 | 42.56 |
| 38 | | 0.4 | 14 | 2.8 | 4 | 40.55 |
| 39 | | 0.4 | 14 | 4.2 | 4 | 37.87 |
| 40 | | 0.4 | 14 | 0 | 4 | 31.91 |
| 41 | | 0.4 | 14 | 2.8 | 9 | 43.14 |
| 42 | | 0.4 | 14 | 1.4 | 9 | 39.82 |
| 43 | | 0.4 | 14 | 4.2 | 9 | 38.46 |
| 44 | | 0.4 | 14 | 0 | 9 | 33.83 |
| 45 | | 0.4 | 14 | 4.2 | 13 | 39.72 |
| 46 | | 0.4 | 14 | 2.8 | 13 | 36.40 |
| 47 | | 0.4 | 14 | 0 | 13 | 33.10 |
| 48 | | 0.4 | 14 | 1.4 | 13 | 35.75 |
| 49 | | 0.4 | 21 | 0 | 0 | 26.95 |
| 50 | | 0.4 | 21 | 1.4 | 0 | 36.27 |
| 51 | | 0.4 | 21 | 4.2 | 0 | 40.90 |
| 52 | | 0.4 | 21 | 2.8 | 0 | 39.58 |
| 53 | | 0.4 | 21 | 1.4 | 4 | 46.86 |
| 54 | | 0.4 | 21 | 2.8 | 4 | 45.51 |
| 55 | | 0.4 | 21 | 0 | 4 | 33.54 |
| 56 | | 0.4 | 21 | 4.2 | 4 | 41.50 |
| 57 | | 0.4 | 21 | 2.8 | 9 | 47.43 |
| 58 | | 0.4 | 21 | 1.4 | 9 | 43.45 |
| 59 | | 0.4 | 21 | 0 | 9 | 36.80 |
| 60 | | 0.4 | 21 | 4.2 | 9 | 42.75 |
| 61 | | 0.4 | 21 | 2.8 | 13 | 44.70 |
| 62 | | 0.4 | 21 | 1.4 | 13 | 41.37 |
| 63 | | 0.4 | 21 | 0 | 13 | 38.72 |
| 64 | | 0.4 | 21 | 4.2 | 13 | 46.00 |
| 65 | | 0.4 | 28 | 1.4 | 0 | 29.97 |
| 66 | | 0.4 | 28 | 4.2 | 0 | 37.94 |
| 67 | | 0.4 | 28 | 2.8 | 0 | 35.29 |
| 68 | | 0.4 | 28 | 0 | 4 | 31.91 |
| 69 | | 0.4 | 28 | 4.2 | 4 | 37.87 |
| 70 | | 0.4 | 28 | 1.4 | 4 | 42.56 |
| 71 | | 0.4 | 28 | 2.8 | 4 | 40.55 |
| 72 | | 0.4 | 28 | 4.2 | 9 | 38.46 |
| 73 | | 0.4 | 28 | 1.4 | 9 | 39.82 |
| 74 | | 0.4 | 28 | 2.8 | 9 | 43.14 |
| 75 | | 0.4 | 28 | 0 | 9 | 33.83 |

**Table 1.** *Cont.*

| No. | Ref. | w/c Ratio per Mass | Curing Time (Days) | Additives (%) | | Compressive Strength (MPa) |
|---|---|---|---|---|---|---|
| | | | | Nanosilica (30–100 nm) (NS) | Microsilica (0.2 μm) (MS) | |
| 76 | | 0.4 | 28 | 4.2 | 13 | 39.72 |
| 77 | | 0.5 | 3 | 0 | 0 | 26.95 |
| 78 | | 0.5 | 3 | 4.2 | 13 | 42.71 |
| 79 | | 0.5 | 7 | 0 | 0 | 16.47 |
| 80 | | 0.5 | 7 | 1.4 | 0 | 21.76 |
| 81 | | 0.5 | 7 | 4.2 | 0 | 25.88 |
| 82 | | 0.5 | 7 | 2.8 | 0 | 25.29 |
| 83 | | 0.5 | 7 | 1.4 | 4 | 31.76 |
| 84 | | 0.5 | 7 | 0 | 4 | 22.35 |
| 85 | | 0.5 | 7 | 4.2 | 4 | 27.65 |
| 86 | | 0.5 | 7 | 2.8 | 4 | 30.59 |
| 87 | | 0.5 | 7 | 4.2 | 9 | 28.82 |
| 88 | | 0.5 | 7 | 0 | 9 | 23.53 |
| 89 | | 0.5 | 7 | 2.8 | 9 | 33.53 |
| 90 | | 0.5 | 7 | 1.4 | 9 | 30.00 |
| 91 | | 0.5 | 7 | 4.2 | 13 | 30.59 |
| 92 | | 0.5 | 7 | 2.8 | 13 | 29.41 |
| 93 | | 0.5 | 7 | 0 | 13 | 26.47 |
| 94 | | 0.5 | 7 | 1.4 | 13 | 28.24 |
| 95 | | 0.5 | 14 | 0 | 0 | 25.03 |
| 96 | | 0.5 | 14 | 1.4 | 0 | 28.06 |
| 97 | | 0.5 | 14 | 4.2 | 0 | 32.89 |
| 98 | | 0.5 | 14 | 2.8 | 0 | 31.66 |
| 99 | | 0.5 | 14 | 1.4 | 4 | 37.17 |
| 100 | | 0.5 | 14 | 2.8 | 4 | 36.04 |
| 101 | | 0.5 | 14 | 0 | 4 | 27.61 |
| 102 | | 0.5 | 14 | 4.2 | 4 | 34.28 |
| 103 | | 0.5 | 14 | 4.2 | 9 | 34.49 |
| 104 | | 0.5 | 14 | 1.4 | 9 | 34.99 |
| 105 | | 0.5 | 14 | 2.8 | 9 | 38.60 |
| 106 | | 0.5 | 14 | 0 | 9 | 30.20 |
| 107 | | 0.5 | 14 | 1.4 | 13 | 32.81 |
| 108 | | 0.5 | 14 | 4.2 | 13 | 37.06 |
| 109 | | 0.5 | 14 | 2.8 | 13 | 35.84 |
| 110 | | 0.5 | 14 | 0 | 13 | 32.18 |
| 111 | | 0.5 | 21 | 0 | 0 | 25.44 |
| 112 | | 0.5 | 21 | 1.4 | 0 | 32.03 |
| 113 | | 0.5 | 21 | 4.2 | 0 | 36.27 |
| 114 | | 0.5 | 21 | 2.8 | 0 | 35.04 |
| 115 | | 0.5 | 21 | 2.8 | 4 | 40.60 |
| 116 | | 0.5 | 21 | 1.4 | 4 | 41.74 |
| 117 | | 0.5 | 21 | 4.2 | 4 | 37.06 |
| 118 | | 0.5 | 21 | 0 | 4 | 31.58 |
| 119 | | 0.5 | 21 | 4.2 | 9 | 37.85 |
| 120 | | 0.5 | 21 | 1.4 | 9 | 38.97 |
| 121 | | 0.5 | 21 | 2.8 | 9 | 43.19 |
| 122 | | 0.5 | 21 | 0 | 9 | 34.77 |
| 123 | | 0.5 | 21 | 1.4 | 13 | 37.39 |
| 124 | | 0.5 | 21 | 0 | 13 | 35.55 |
| 125 | | 0.5 | 21 | 4.2 | 13 | 41.62 |
| 126 | | 0.5 | 21 | 2.8 | 13 | 40.41 |
| 127 | | 0.5 | 28 | 0 | 0 | 28.57 |
| 128 | | 0.5 | 28 | 1.4 | 0 | 36.31 |
| 129 | | 0.5 | 28 | 4.2 | 0 | 40.48 |
| 130 | | 0.5 | 28 | 2.8 | 0 | 38.69 |

**Table 1.** *Cont.*

| No. | Ref. | w/c Ratio per Mass | Curing Time (Days) | Additives (%) | | Compressive Strength (MPa) |
|---|---|---|---|---|---|---|
| | | | | Nanosilica (30–100 nm) (NS) | Microsilica (0.2 μm) (MS) | |
| 131 | | 0.5 | 28 | 1.4 | 4 | 45.24 |
| 132 | | 0.5 | 28 | 2.8 | 4 | 44.05 |
| 133 | | 0.5 | 28 | 4.2 | 4 | 42.26 |
| 134 | | 0.5 | 28 | 0 | 4 | 33.33 |
| 135 | | 0.5 | 28 | 2.8 | 9 | 47.62 |
| 136 | | 0.5 | 28 | 1.4 | 9 | 43.45 |
| 137 | | 0.5 | 28 | 4.2 | 9 | 42.26 |
| 138 | | 0.5 | 28 | 0 | 9 | 37.50 |
| 139 | | 0.5 | 28 | 0 | 13 | 38.69 |
| 140 | | 0.5 | 28 | 4.2 | 13 | 45.24 |
| 141 | | 0.5 | 28 | 1.4 | 13 | 41.67 |
| 142 | | 0.5 | 28 | 2.8 | 13 | 44.05 |
| 143 | | 0.84 | 3 | 5 | 5 | 15.00 |
| 144 | | 0.84 | 3 | 0 | 5 | 14.00 |
| 145 | | 0.84 | 7 | 0 | 5 | 15.00 |
| 146 | [34] | 0.84 | 7 | 5 | 5 | 17.00 |
| 147 | | 0.84 | 14 | 5 | 5 | 20.00 |
| 148 | | 0.84 | 14 | 0 | 5 | 18.00 |
| 149 | | 0.84 | 28 | 5 | 5 | 24.00 |
| 150 | | 0.84 | 28 | 0 | 5 | 20.00 |
| 151 | | 0.6 | 7 | 2 | 5 | 34.09 |
| 152 | | 0.6 | 7 | 3 | 5 | 31.79 |
| 153 | | 0.6 | 28 | 1 | 5 | 43.69 |
| 154 | | 0.6 | 28 | 2 | 5 | 44.47 |
| 155 | [35] | 0.6 | 28 | 0 | 5 | 41.04 |
| 156 | | 0.6 | 28 | 3 | 5 | 42.81 |
| 157 | | 0.6 | 90 | 0 | 5 | 50.03 |
| 158 | | 0.6 | 90 | 2 | 5 | 53.68 |
| 159 | | 0.6 | 90 | 1 | 5 | 51.04 |
| 160 | | 0.4 | 3 | 0 | 0 | 22.80 |
| 161 | | 0.4 | 3 | 2 | 0 | 22.60 |
| 162 | | 0.4 | 3 | 1.5 | 0 | 24.00 |
| 163 | | 0.4 | 3 | 0 | 10 | 19.90 |
| 164 | | 0.4 | 3 | 2 | 10 | 18.80 |
| 165 | | 0.4 | 7 | 2 | 0 | 27.10 |
| 166 | [36] | 0.4 | 7 | 1.5 | 0 | 28.80 |
| 167 | | 0.4 | 7 | 0 | 15 | 25.80 |
| 168 | | 0.4 | 7 | 2 | 10 | 26.10 |
| 169 | | 0.4 | 7 | 0 | 10 | 29.40 |
| 170 | | 0.4 | 28 | 0 | 0 | 34.50 |
| 171 | | 0.4 | 28 | 2 | 0 | 32.50 |
| 172 | | 0.4 | 28 | 1.5 | 0 | 37.90 |
| 173 | | 0.4 | 28 | 0 | 15 | 42.60 |
| 174 | | 0.5 | 3 | 1 | 0 | 29.21 |
| 175 | | 0.5 | 3 | 2.5 | 0 | 31.31 |
| 176 | | 0.5 | 3 | 2 | 0 | 31.55 |
| 177 | | 0.5 | 3 | 1.5 | 0 | 30.61 |
| 178 | [37] | 0.5 | 3 | 0 | 0 | 25.00 |
| 179 | | 0.5 | 3 | 0 | 30 | 28.27 |
| 180 | | 0.5 | 3 | 2 | 30 | 35.52 |
| 181 | | 0.5 | 3 | 0 | 20 | 27.10 |
| 182 | | 0.5 | 3 | 2 | 20 | 37.16 |
| 183 | | 0.5 | 3 | 2 | 10 | 32.25 |

**Table 1.** *Cont.*

| No. | Ref. | w/c Ratio per Mass | Curing Time (Days) | Additives (%) | | Compressive Strength (MPa) |
|---|---|---|---|---|---|---|
| | | | | Nanosilica (30–100 nm) (NS) | Microsilica (0.2 µm) (MS) | |
| 184 | | 0.5 | 3 | 0 | 10 | 25.70 |
| 185 | | 0.5 | 7 | 0 | 0 | 31.08 |
| 186 | | 0.5 | 7 | 0 | 40 | 35.29 |
| 187 | | 0.5 | 7 | 2 | 40 | 42.07 |
| 188 | | 0.5 | 7 | 2 | 30 | 43.71 |
| 189 | | 0.5 | 7 | 0 | 30 | 36.22 |
| 190 | | 0.5 | 7 | 2 | 20 | 44.18 |
| 191 | | 0.5 | 7 | 0 | 20 | 34.82 |
| 192 | | 0.5 | 7 | 0 | 10 | 33.18 |
| 193 | | 0.5 | 7 | 2 | 10 | 39.26 |
| 194 | [37] | 0.5 | 28 | 1 | 0 | 51.66 |
| 195 | | 0.5 | 28 | 0 | 0 | 44.88 |
| 196 | | 0.5 | 28 | 2.5 | 0 | 54.47 |
| 197 | | 0.5 | 28 | 2 | 0 | 54.70 |
| 198 | | 0.5 | 28 | 2 | 40 | 62.19 |
| 199 | | 0.5 | 28 | 0 | 40 | 52.36 |
| 200 | | 0.5 | 28 | 2 | 30 | 64.06 |
| 201 | | 0.5 | 28 | 0 | 30 | 53.30 |
| 202 | | 0.5 | 28 | 0 | 20 | 51.43 |
| 203 | | 0.5 | 28 | 2 | 20 | 66.87 |
| 204 | | 0.5 | 28 | 2 | 10 | 59.85 |
| 205 | | 0.5 | 28 | 0 | 10 | 48.85 |
| **No. of Data = 205** | | **Ranged between 0.4 and 0.84** | **Varied between 3 and 90 Days** | **Ranged between 0 and 15%** | **Ranged between 0 and 40%** | **Varied between 14 and 67 MPa** |

*2.2. Statistical Evaluation*

Statistical analysis was performed in the current section to demonstrate the association between input parameters and paste compressive strength. In this respect, all variables considered, including (i) w/c, (ii) curing time, (iii) NS content, and (iv) MS content, were collected and evaluated with compressive strength, and statistical functions such as standard deviation, skewness, kurtosis, and variance were calculated to demonstrate the distribution of each variable with compressive strength. Concerning the kurtosis parameter, the higher negative value shows shorter distribution tails than the normal distribution, whereas the longer tails represent the positive value. About the skewness, the negative value indicates the long left tail, and the positive value is the right tail. Below is ample information on each vector considered as an input parameter:

**(i) Water/cement ratio (w/c)**

According to the comprehensive data collected from the previous analyses, the w/c of cement paste mixes adjusted with NS and MS varied from 0.4 to 0.84 with a median of 0.5. Statistical evaluation for the parameters such as variance, standard deviation, skewness, and kurtosis were 0.0119, 0.1091, 0.2.082, and 4.5212, respectively. Additionally, Figure 2a shows a difference between the compressive intensity and the w/c ratio—the histogram of modified paste mixtures with NS and MS. Figure 2b indicates that, with an improvement in the w/c of 0.55%, the compressive strength of the cement paste changed with NS and MS decreased substantially.

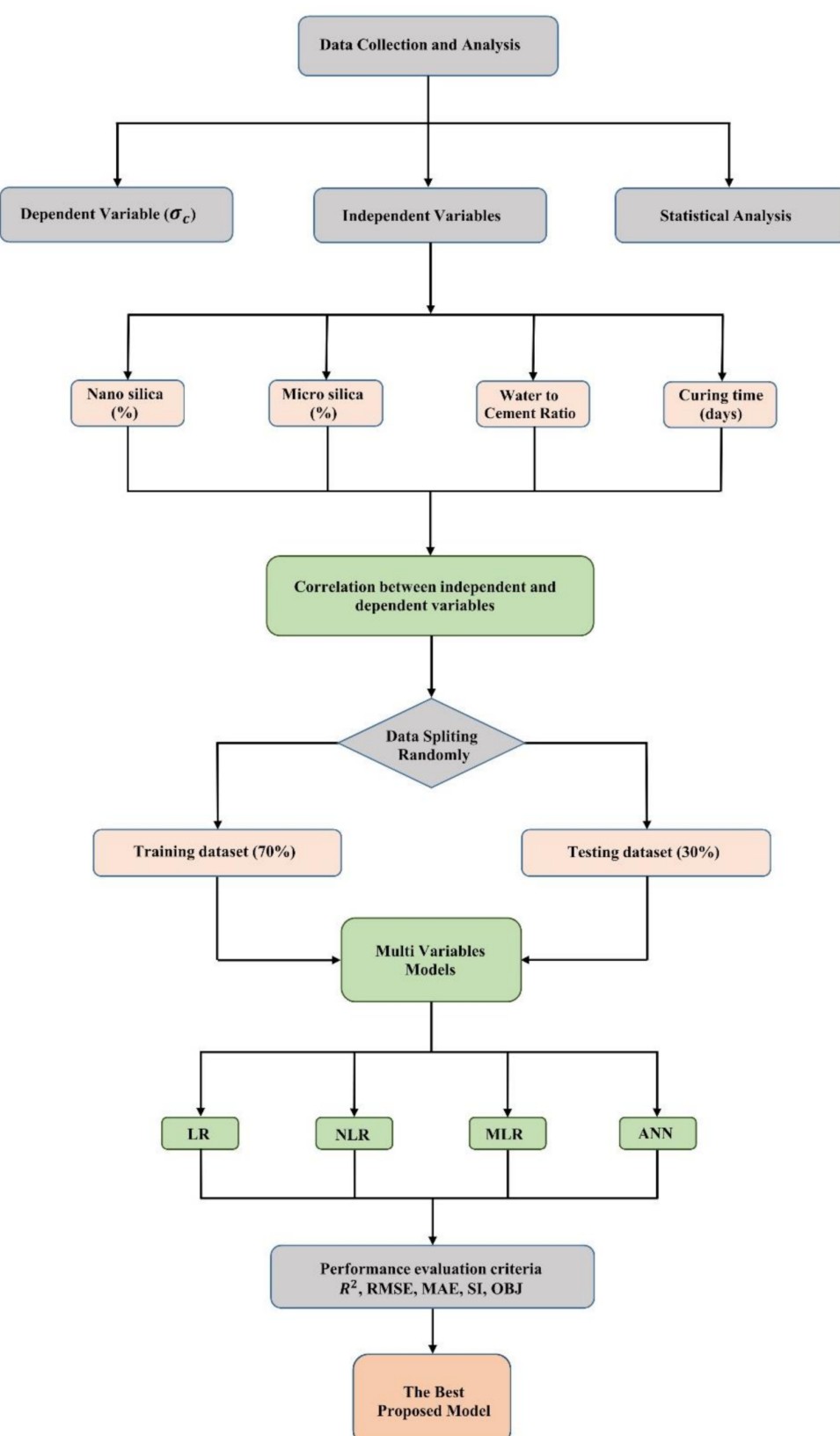

**Figure 1.** The flowchart followed in this study for cement paste modified with NS and MS.

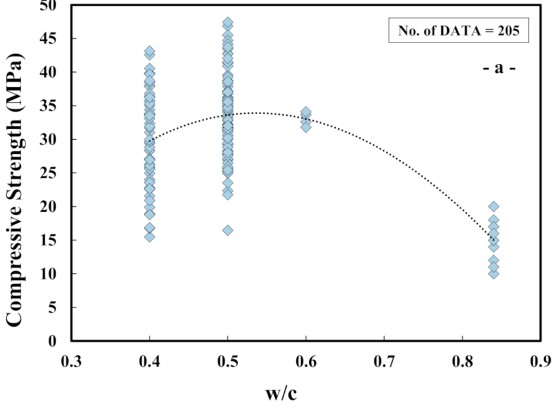
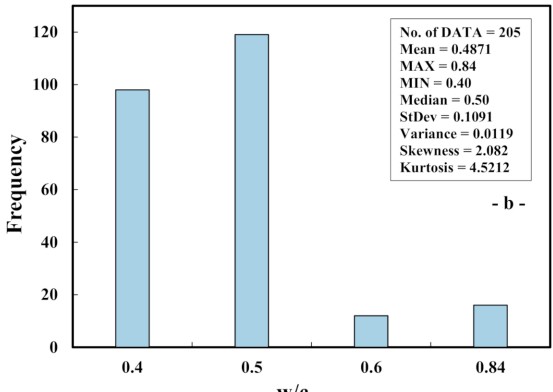

**Figure 2.** (**a**) The variation between the compressive strength and w/c and (**b**) histogram of w/c of cement paste modified with NS and MS.

**(ii)   Curing time**

To facilitate the hydration procedure, the curing duration should be extended enough to obtain an acceptable early and hardened age compressive strength. Thus, based on the literature data, the curing period range of NS and MS modified paste began from 3 days to 90 days, with a median of 14 days. Based on skewness value, the curing time had a higher value compared to the w/c ratio skewness, which was 28.27%. Furthermore, the kurtosis value of curing time was 63.68% higher than the w/c ratio kurtosis. The association between compressive strength and curing time with the NS and MS-altered paste mixture is indicated in Figure 3.

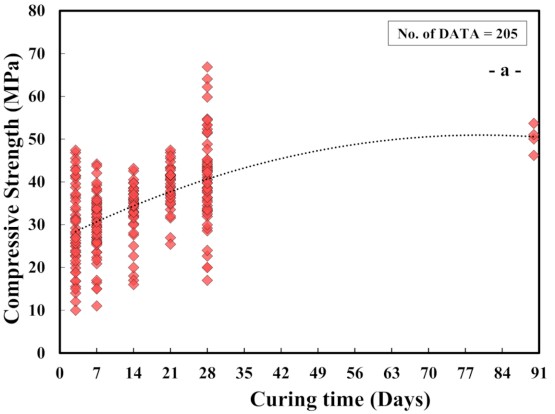
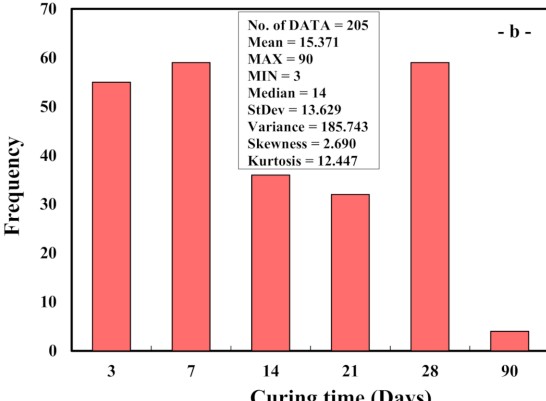

**Figure 3.** (**a**) The variation between the compressive strength and curing time and (**b**) histogram of curing time of cement paste modified with NS and MS.

**(iii)   NS content**

The NS utilized in the mixing proportions had a particle diameter of less than 50 nm, a surface-to-volume ratio of between 50 and 200 $m^2/g$, and a purity of more than 99 percent, according to the examination of the gathered dataset from the literature. Between 0% and 15% of NS was employed in paste mixes to replace the cement weight, with a median of 1.5 percent. In comparison, the standard deviation, uncertainty, skewness, and kurtosis were 2.47, 6.1, 2.737, and 11.11, respectively. The difference in compressive strength, NS histogram analysis, and histogram analysis are indicated in Figure 4b. As seen in Figure 4a, there was a weak association between compressive strength and NS content of more than 4%.

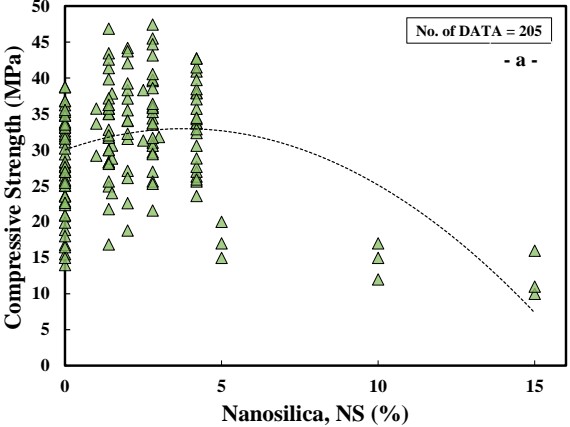
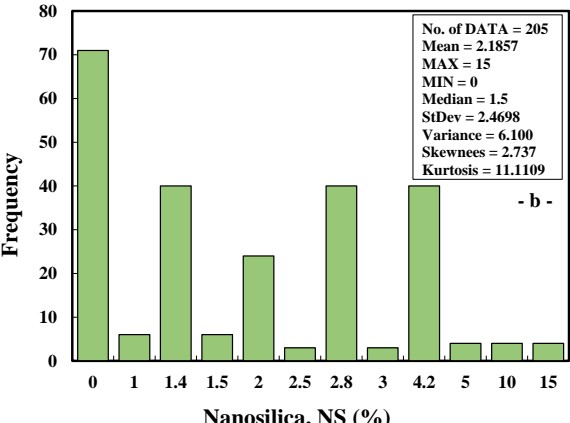

**Figure 4.** (**a**) The variation between the compressive strength and nanosilica (NS) and (**b**) histogram of nanosilica (NS) of cement paste.

**(iv) MS content**

The MS utilized in the mix had a particle diameter of less than 50 nm, a surface-to-volume ratio of 10–20 m$^2$/g, and purity of greater than 99%, according to the dataset derived from the literature. The minimum and maximum proportions of MS used in paste mixtures were between 0% and 40%, replacing the cement weight with a median of 5%. Based on the Kurtosis value, MS had a lower value than NS, almost 50%. Furthermore, the skewness value of MS was 28% less than NS skewness. The difference in compressive strength between MS material and the histogram analysis is seen in Figure 5. As indicated in Figure 5a, there was a slowly reducing relationship between compressive strength and MS content of more than 30%.

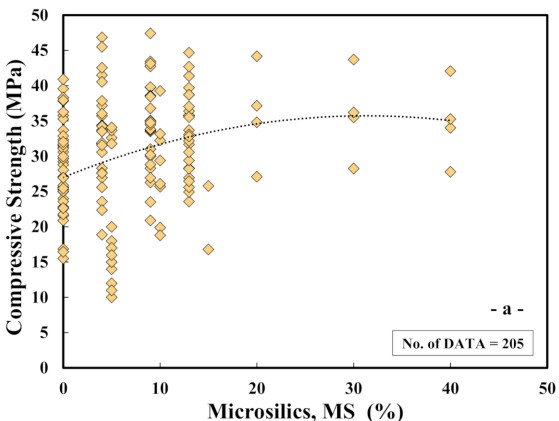
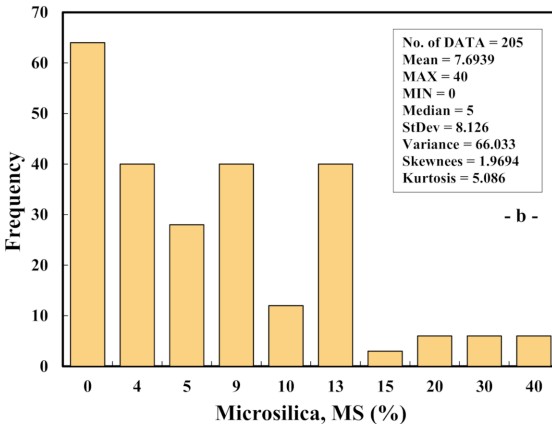

**Figure 5.** (**a**) The variation between the compressive strength and microsilica (MS) and (**b**) histogram of microsilica (MS) of cement paste.

**(v) Compressive strength**

The compressive strength of paste mixtures changed with NS and MS varied from 10 to 66.87 MPa. A median of 34.76 MPa, a standard deviation of 9.80 MPa, and a variance of 96.067 MPa were recorded according to the total data recorded in Table 1. From the overall collected results, 73% had compressive strengths ranging from 10 to 40 MPa, and 27% had compressive strengths ranging from 40 to 66.87 MPa (Table 1).

*2.3. Modeling*

Four separate models were suggested to test the effect of the different mixture proportions referred to above on the compressive strength of NS and MS-adjusted cement paste. The proposed models were used in this analysis to estimate the compressive strength of

cement paste. It needed specific criteria to choose the best model that gave closer estimated results to the measured compressive strength. The following calculation parameters were used to compare the predictions of various models. The most corrected model statistically had a smaller percentage error between the actual and predicted results and had a lower RMSE and SI OBJ and a higher $R^2$.

### 2.3.1. Linear Regression Model (LR)

Linear regression tries to fit the linear equation to the measurable data to predict the relationship between the two variables. The LR model is the most common approach to predicting the compressive strength of concrete [38]. The equation for a linear regression line is Equation (1).

$$Y = a + bX \tag{1}$$

a and b are the model parameters, X is the explanatory variable, and Y is the dependent variable (Equation (1)). Several ingredients and variables that affected paste mixtures' compressive strength adjusted with NS and MS, such as curing time (t), water-cement ratio (w/c), and other blend proportions, could be included in the equation above. Equation (2) was recommended to incorporate all parameters and quantities that may have affected compressive strength in order to provide more accurate and empirical observations.

w/c is the ratio of water to cement, t is the curing time, NS is the content of nanosilica (%), and MS is microsilica (%). In addition, the model parameters are a, b, c, d, and e. The suggested Equation (2), since all variables can be adapted linearly, can be used to an extent for Equation (1). While all the different compressive strength variables can influence and interfere with one another, this may not be important in all cases. The model must also constantly be updated to accurately measure the compressive strength [39,40].

$$\sigma_c = a(w/c) + b(t) + c(NS) + d(MS) + e \tag{2}$$

### 2.3.2. Nonlinear Regression Model (NLR)

The following Equation (3) can be used to develop a nonlinear regression model in a general form [40–42]. Equation (3) was used to approximate the compressive strength of paste mixtures improved with NS and MS to describe the interrelationship between the various variables in Equations (1) and (2).

$$\sigma_c = a(w/c)^b + c(t)^d + e(NS)^f + g(MS)^h + i \tag{3}$$

where w/c is the water to cement ratio, t is the curing time (days), NS is the nanosilica content (%), and MS is the microsilica content (%). Furthermore, the model parameters a, b, c, d, e, f, g, h, and i were determined using the least square method.

### 2.3.3. Multi-Logistic Regression Model (MLR)

The MLR, a regression algorithm, can be used where the predictable variable has a parameter greater than two phases. The MLR is a mathematical method that is close to multiple linear regression. The equation can also find the difference between a predictor variable and the independent variables of Equation (4).

$$\sigma_c = a(w/c)^b (t)^C (NS)^d (MS)^e \tag{4}$$

Equation (14) has a drawback in that it cannot be used to estimate the compressive strength of paste without the presence of NS and MS. As a result, the NS and MS content in this model should be larger than zero (the constraint of Equation (4) is NS and MS content >0%). The least-square approach was also used to find the parameters of the models a, b, c, d, and e and model variables.

### 2.3.4. Artificial Neural Network Model (ANN)

A multilayer perceptron is a neural network connecting multiple layers in a directed graph, which means that the signal path through the nodes only goes one way. Apart from the input nodes, each node has a nonlinear activation function. Since there are multiple layers of neurons, ANN is a deep learning technique. ANN is widely used to solve computational neuroscience and parallel distributed processing research problems. The multilayer perceptron is an ANN feedforward class. The ANN procedure consists of three main parts: the input layer (variables), hidden layers (neurons and additional layers), and the output layer (desired output result). In this study, the main input variables of Equations (5) and (6) involved water-cement ratio, curing days, and additives (nanosilica and microsilica).

$$\beta_n = a_n(w/c) + b_n(t) + c_n(NS) + d_n(MS) + e_n \tag{5}$$

$$\sigma_c = \frac{Node_1}{1 + e^{-\beta 1}} + \frac{Node_2}{1 + e^{-\beta 2}} + \ldots + \frac{Node_n}{1 + e^{-\beta n}} + Treshold \tag{6}$$

where w/c is the water to cement ratio (%), t is the curing time (days), NS is the nanosilica content (%), and MS is the microsilica content (%). All the β values that depend on the mentioned values by linear Equations (5) and (6) directly given by the neural learning machine (Weka), n is several neurons that are defined with the best of trial and error by the same machine which provides the most value of $R^2$. Furthermore, $node_1$, $node_2$ ... $node_n$, and all the other model parameters a, b, c, d, e, and threshold could be determined by WEKA software directly with the same trial effort.

### 2.3.5. Criteria for Evaluation of Models

Various output parameters, including the coefficient of determination ($R^2$), Root Mean Squared Error (RMSE), Mean Absolute Error (MAE), Scatter Index (SI), and OBJ, which were specified, were used to test and evaluate the efficiency of the proposed models.

$$R^2 = \left( \frac{\sum_{p=1}^{p}(t_p - t')\left(y_p - y'\right)}{\sqrt{\left[\sum_{p=1}^{p}(t_p - t')^2\right]\left[\sum_{p=1}^{p}\left(y_p - y'\right)^2\right]}} \right)^2 \tag{7}$$

$$RMSE = \sqrt{\frac{\sum_{p=1}^{p}\left(y_p - t_p\right)^2}{p}} \tag{8}$$

$$MAE = \frac{\sum_{p=1}^{p}\left|\left(y_p - t_p\right)\right|}{p} \tag{9}$$

$$SI = \frac{RMSE}{t'} \tag{10}$$

$$OBJ = \left(\frac{n_{tr}}{n_{all}} * \frac{RMSE_{tr} + MAE_{tr}}{R_{tr}^2 + 1}\right) + \left(\frac{n_{tst}}{n_{all}} * \frac{RMSE_{tst} + MAE_{tst}}{R_{tst}^2 + 1}\right) \tag{11}$$

where $y_p$ and $t_p$ are the predicted and measured values of the path trend, correspondingly, and $t'$ and $y'$ are the averages of the measured and predicted values, respectively. n is the number of trends (collected data) in the relevant dataset, and $_{tr}$, $_{tst}$ and $_{all}$ are the training, testing, and all datasets, respectively.

However, for the $R^2$ value, the closer the numerical value of $R^2$ to the value of 1, the stronger and more accurate the obtained model. When it came to the SI parameter, a model had poor performance when it was >0.3, fair performance when it was 0.2 SI 0.3, good performance when it was 0.1 SI 0.2, and excellent performance when it was <0.1 [38,43].

Moreover, the OBJ function was used as a critical output parameter in Equation (11) to determine the utility of the proposed models.

### 2.3.6. Sensitivity of Parameters

In order to understand the magnitude of the effect of each parameter on the compressive strength in the selected superior model, the following proposed method was used. Each time one of the parameters was removed from the superior model, values were obtained for each of the three criteria such as $R^2$, RMSE, and MAE. Finally, by comparing the results, the most change would mean the highest sensitivity of the ordering parameter. In the case of $R^2$, the lower value and, in the case of RMSE and MAE, the highest value would represent the most effective parameter.

## 3. Results

### 3.1. Predicted and Measured Compressive Strength Relationships

### 3.1.1. The LR Model

Figure 6 displays the relationship between the expected and real compressive strength of paste mixtures adjusted with NS and MS for training and testing datasets, respectively. According to the model parameters, the w/c ratio and NS material had a major effect on the compressive strength of the paste adjusted with NS and MS. The weight of each parameter on the compressive strength of paste adjusted with NS and MS was calculated for the current model by the least square approach in Excel [44,45]. The equation for the LR model could be written as follows for various weight parameters (Equation (12)).

$$\sigma_c = -22.17 \, ^w/_c + 0.5t + 1.32NS + 0.47MS + 31.23 \tag{12}$$

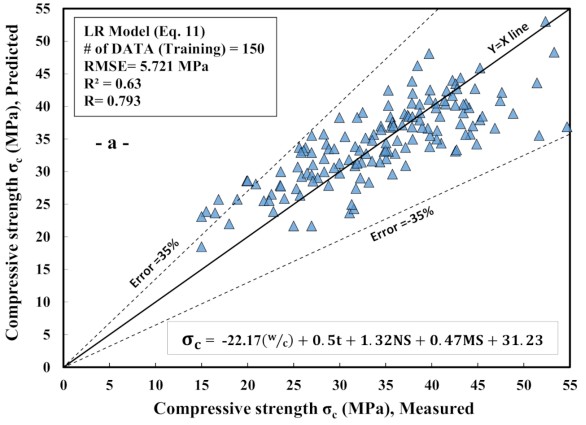
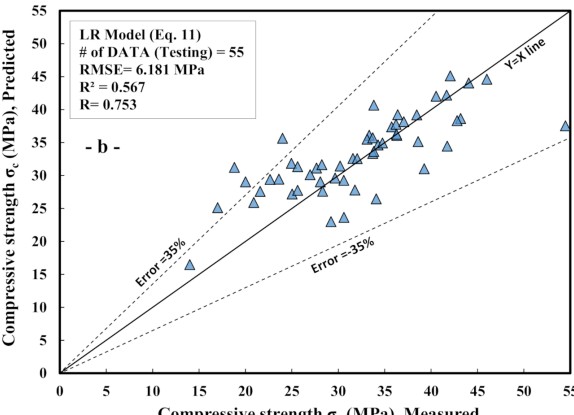

**Figure 6.** Comparison between the measured and predicted compressive strength of cement paste with nanosilica (NS) and microsilica (MS) using Linear Regression model (LR). (**a**) Training dataset and (**b**) testing dataset.

As can be seen from the equation above, the w/c ratio had the greatest effect on the compressive strength reduction in all factors. This may correspond to the experimental findings reported in the literature [46,47]. The assessment parameters for this model, such as the $R^2$, MAE, and RMSE, were 0.63, 4.44 MPa, and 5.72 MPa, respectively. Moreover, the OBJ and SI values for the current model were 4.56 and 0.16 for the training dataset.

### 3.1.2. NLR Model

Figure 7 displays the actual compressive strength versus the predicted compressive strength based on experimental studies of paste mixtures altered with NS and MS for training and testing datasets accordingly. This model showed that the w/c ratio was the most significant parameter that influenced the compressive strength of the paste mixtures. Several previous experiments demonstrated this, in which decreases in the w/c ratio

substantially increased the compressive strength of the cement paste mixtures [6,7,48,49]. The following is the suggested equation for the NLR model with various vector parameters (Equation (13)).

$$\sigma c = 35(\text{w/c})^{-0.176} + 8.635t^{0.37} + 4.65\text{NS}^{0.2} + 0.33\text{MS}^{1.1} - 35 \tag{13}$$

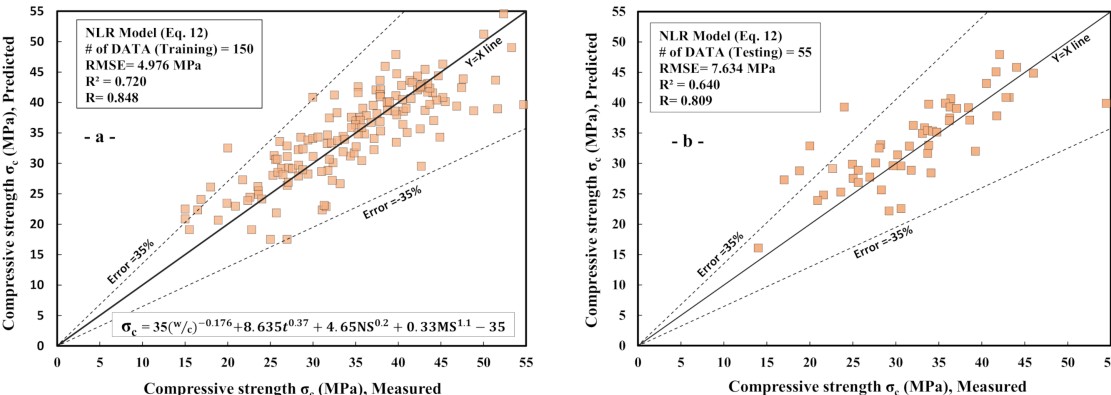

**Figure 7.** Comparison between the measured and predicted compressive strength of cement paste with nanosilica (NS) and microsilica (MS) using Non-Linear Regression model (NLR). (**a**) Training dataset and (**b**) testing dataset.

The $R^2$, RMSE, and MAE evaluation parameters for this model were 0.72, 4.98 MPa, and 3.81 MPa. Furthermore, the current OBJ of the model and SI values for the training dataset were 3.74 and 0.14, respectively.

### 3.1.3. Multi-Logistic Regression Model (MLR)

Figure 8 demonstrates the relationship between the expected compressive strength versus the actual compressive strength based on experimental studies with paste mixtures adjusted with NS and MS for training and testing datasets, collectively. The curing time was the most effective aspect that influenced the paste mixtures' compressive strength adjusted with NS and MS compared to other models. Additionally, among additives, the effect of NS was greater than MS. Equation (14) reflects the established model for the MLR model with different control factors.

$$\sigma c = 18.73(\text{w/c})^{-0.121}\, t^{0.227}\, \text{NS}^{0.011}\, \text{MS}^{0.008} \tag{14}$$

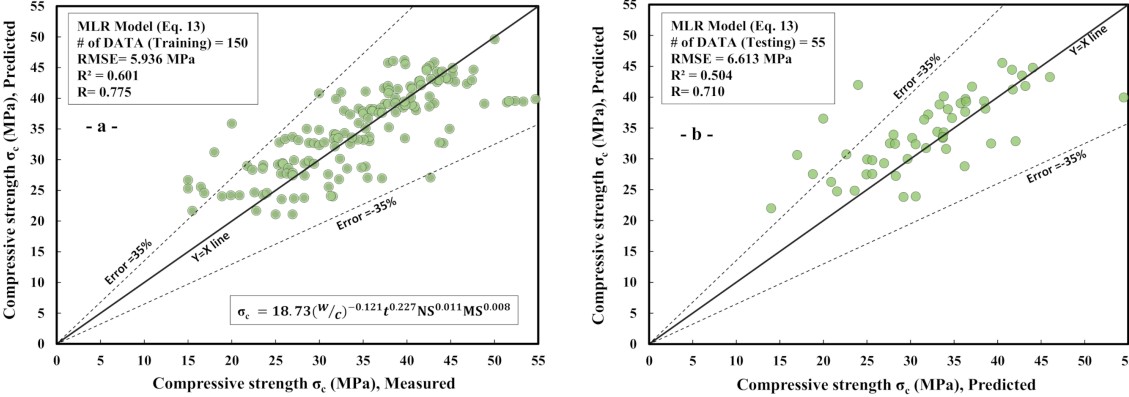

**Figure 8.** Comparison between the measured and predicted compressive strength of Cement paste with nanosilica (NS) and microsilica (MS) using Multi-Linear regression model (MLR). (**a**) Training dataset and (**b**) testing dataset.

The $R^2$, RMSE, and MAE evaluation parameters for this model were 0.60, 5.94 MPa, and 4.35 MPa. Furthermore, the current OBJ of the model and SI values for the training dataset were 4.70 and 0.169, respectively.

### 3.1.4. Artificial Neural Network Model (ANN)

Figure 9 displays the predicted compressive strength versus actual compressive strength based on experimental studies of paste mixtures altered with NS and MS for training and testing datasets accordingly. As shown by this model on the basis of Figure 10, the curing time was the most significant parameter that influenced the compressive strength of the paste mixtures. The following (Equations (15) and (16)) are the suggested equations for the ANN model based on Equations (15) and (6) with various vector parameters obtained by WEKA machine learning algorithms. These parameters were collected in Table 2 and applied to the following matrix (Equation (15)) to obtain the value of β1–β14. It should be noted that through the 28 trials conducted in WEKA with various numbers of hidden layers and neurons (Table 3), the case with 2 hidden layers and 21 neurons (14, 7) was the best one with the highest $R^2$ value and the lowest amount of RMSE and MAE compared to another, but due to equation complexity, the case with 1 hidden layer and 14 neurons was chosen, which had the best conditions through the one-layer trials (Figures 11 and 12).

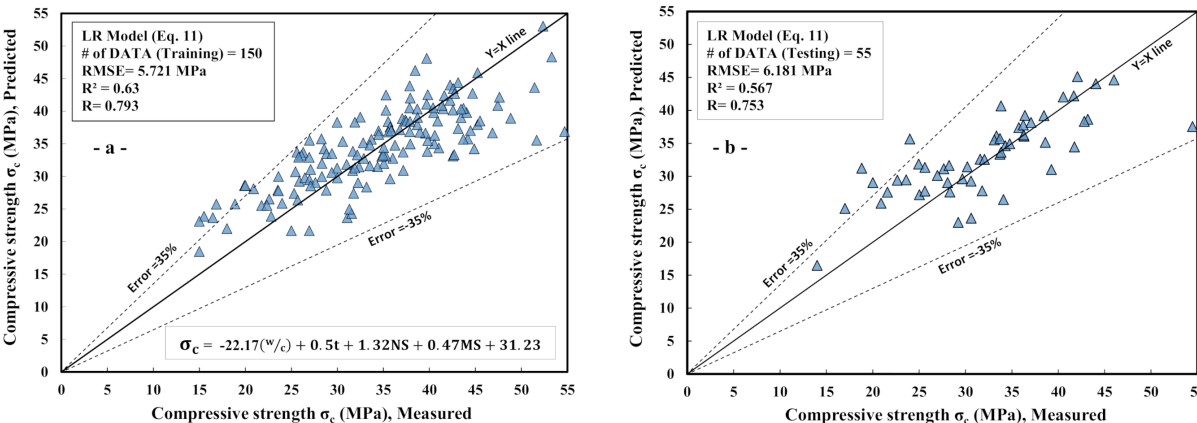

**Figure 9.** Comparison between the measured and predicted compressive strength of cement paste with nanosilica (NS) and microsilica (MS) using Linear Regression model (LR). (**a**) Training dat taset and (**b**) testing dataset.

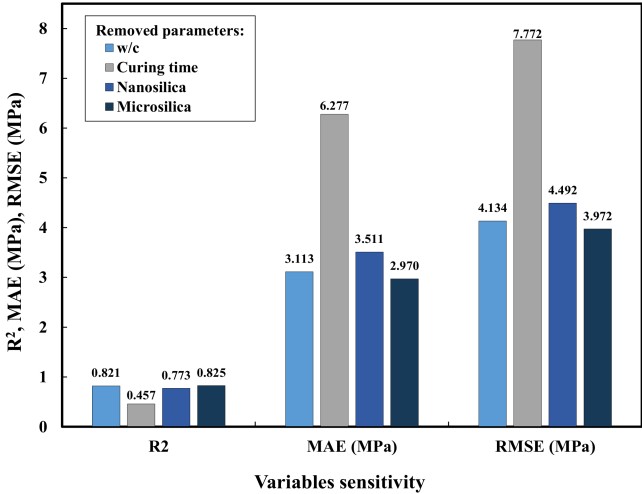

**Figure 10.** Sensitivity comparison of the variables on the compressive strength of cement paste with nanosilica (NS) and microsilica (MS) using ANN model.

**Table 2.** The parameters for compressive strength using the ANN model based on trial no.14 in Table 2 and using them in (Equations (15) and (16)).

| | LM No: | 1 | 2 | 3 | 4 | 5 | 6 | 7 | 8 | 9 | 10 | 11 | 12 | 13 | 14 |
|---|---|---|---|---|---|---|---|---|---|---|---|---|---|---|---|
| Model Parameters | a | 1.71 | −6.42 | 0.08 | 6.84 | 0.28 | 4.02 | 5.35 | −16.8 | 5.89 | −2.73 | 10.73 | 4.25 | 5.83 | 2.08 |
| | b | 6.88 | 7.41 | 2.46 | 3.47 | 12.64 | −1.56 | −6.63 | −2.13 | 2.27 | 7.26 | −1.29 | −0.26 | −2.86 | 1.70 |
| | c | −10.2 | 0.98 | −4.43 | 6.14 | −4.81 | 5.09 | −3.40 | −1.08 | 14.48 | −0.58 | −1.64 | 0.97 | 7.04 | 3.92 |
| | d | 4.83 | −0.67 | −0.83 | −2.05 | −2.78 | 6.71 | 4.56 | 8.22 | −4.98 | 3.00 | 1.33 | 0.03 | 6.22 | −2.30 |
| | e | −3.28 | 0.07 | −5.60 | 5.85 | −3.14 | 0.87 | −6.99 | −10.6 | −7.96 | 2.13 | 6.91 | −2.37 | 1.56 | −2.58 |
| Nodes | | 0.80 | 1.58 | −3.90 | 0.75 | 1.99 | 2.08 | −1.79 | 0.73 | 3.28 | −0.73 | 1.46 | 0.86 | −1.64 | −5.93 |
| Threshold | | | | | | | | −1.75 | | | | | | | |

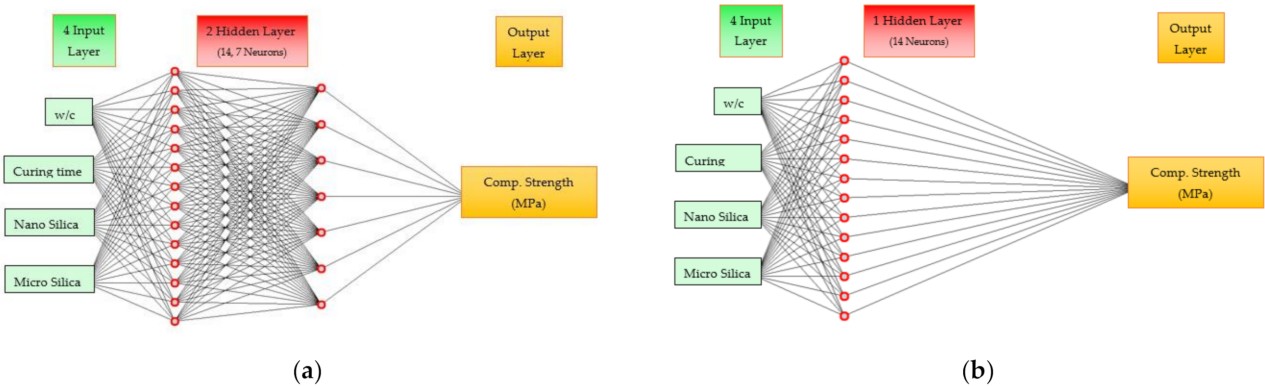

(**a**)          (**b**)

**Figure 11.** The Optimal Architecture of Neural Network Model; (**a**) 14 and 7 neurons; (**b**) 14 neurons.

$$\sigma_c = \frac{0.8}{1 + e^{-\beta 1}} + \frac{1.58}{1 + e^{-\beta 2}} + \ldots + \frac{-5.93}{1 + e^{-\beta 14}} - 1.75 \tag{16}$$

The $R^2$, RMSE, and MAE evaluation parameters for this model were 0.899, 3.01 MPa, and 2.19 MPa, respectively. Furthermore, the current OBJ of the model and SI values for the training dataset were 2.00 and 0.09, respectively.

$$\begin{bmatrix} 1.71 & 6.88 & -10.2 & 4.83 & -3.28 \\ -6.42 & 7.41 & 0.98 & -0.67 & 0.08 \\ 0.08 & 2.46 & -4.43 & -0.83 & -5.6 \\ 6.84 & 3.47 & 6.14 & -2.05 & 5.85 \\ 0.28 & 12.64 & -4.81 & -2.78 & -3.14 \\ 4.02 & -1.56 & 5.09 & 6.71 & 0.87 \\ 5.35 & -6.63 & -3.4 & 4.56 & -6.99 \\ -16.8 & -2.13 & -1.08 & 8.22 & -10.6 \\ 5.89 & 2.27 & 14.48 & -4.98 & -7.96 \\ -2.73 & 7.26 & -0.58 & 3 & 2.13 \\ 10.73 & -1.29 & -1.64 & 1.33 & 6.91 \\ 4.25 & -0.26 & 0.97 & 0.03 & -2.37 \\ 5.83 & -2.86 & 7.04 & 6.22 & 1.56 \\ 2.08 & 1.7 & 3.92 & -2.3 & -2.58 \end{bmatrix} * \begin{bmatrix} \frac{w}{c} \\ t \\ NS \\ MS \\ 1 \end{bmatrix} = \begin{bmatrix} \beta_1 \\ \beta_2 \\ \beta_3 \\ \beta_4 \\ \beta_5 \\ \beta_6 \\ \beta_7 \\ \beta_8 \\ \beta_8 \\ \beta_{10} \\ \beta_{11} \\ \beta_{12} \\ \beta_{13} \\ \beta_{14} \end{bmatrix} \tag{15}$$

**Table 3.** Choosing the best-hidden layer and neurons for the ANN model.

| # of Layers | # of Neuron | # of Neurons for Each Layer * | R ** | MAE (MPa) | RMSE (MPa) |
|---|---|---|---|---|---|
| 1 | 2 | 2 | 0.862 | 4.019 | 5.048 |
| 1 | 3 | 3 | 0.883 | 3.729 | 4.675 |
| 1 | 4 | 4 | 0.897 | 3.467 | 4.416 |
| 1 | 5 | 5 | 0.921 | 3.1 | 3.908 |
| 1 | 6 | 6 | 0.928 | 2.715 | 3.552 |
| 1 | 7 | 7 | 0.929 | 2.623 | 3.515 |
| 1 | 8 | 8 | 0.94 | 2.452 | 3.29 |
| 1 | 9 | 9 | 0.937 | 2.54 | 3.299 |
| 1 | 10 | 10 | 0.943 | 2.395 | 3.149 |
| 1 | 12 | 12 | 0.945 | 2.392 | 3.14 |
| **1** | **14** | **14** | **0.948** | **2.19** | **3.005** |
| 1 | 15 | 15 | 0.944 | 2.443 | 3.122 |
| 2 | 4 | 2 + 2 | 0.857 | 3.862 | 4.869 |
| 2 | 6 | 2 + 4 | 0.857 | 3.851 | 4.863 |
| 2 | 8 | 4 + 4 | 0.901 | 3.286 | 4.202 |
| 2 | 12 | 4 + 8 | 0.942 | 2.383 | 3.158 |
| 2 | 14 | 6 + 8 | 0.95 | 2.143 | 2.936 |
| 2 | 15 | 6 + 9 | 0.931 | 2.557 | 3.422 |
| 2 | 16 | 6 + 10 | 0.939 | 2.436 | 3.269 |
| 2 | 16 | 2 + 14 | 0.889 | 3.46 | 4.317 |
| 2 | 15 | 5 + 10 | 0.932 | 2.612 | 3.408 |
| 2 | 12 | 8 + 4 | 0.956 | 2.048 | 2.771 |
| 2 | 16 | 8 + 8 | 0.962 | 1.815 | 2.552 |
| 2 | 18 | 9 + 9 | 0.964 | 1.863 | 2.58 |
| 2 | 18 | 12 + 6 | 0.961 | 1.878 | 2.627 |
| **2** | **21** | **14 + 7** | **0.977** | **1.392** | **2.1** |
| 3 | 12 | 2 + 4 + 6 | 0.857 | 3.857 | 4.865 |
| 3 | 18 | 3 + 6 + 9 | 0.92 | 2.883 | 3.702 |

* For example, 2 + 4 indicates that there are 2 neurons in the first layer and 4 neurons in the second layer. ** R is the correlation coefficient which the square of that is coefficient of determination ($R^2$).

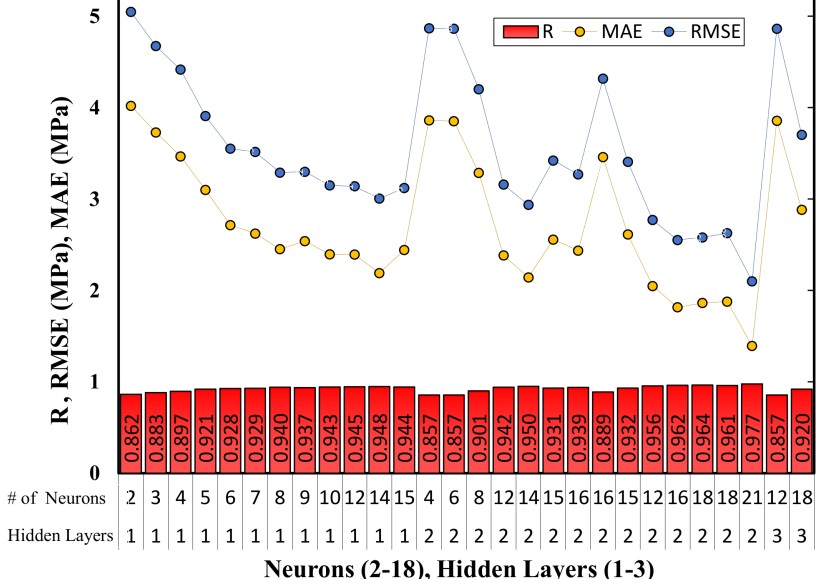

**Figure 12.** Statistical assessments to evaluate the artificial neural network model using a training dataset.

### 3.1.5. Comparison between Developed Models

As stated earlier, five distinct statistical techniques were used to evaluate the suggested models' efficiency: RMSE, SI MAE, $R^2$, and OBJ. The ANN model had a higher $R^2$ with

lower RMSE and MAE values among the four different models compared to the LR, NLR, and MLR models (Figure 13). Additionally, Figure 14 presents the SI and OBJ together to compare the model estimations of the compressive strength of paste mixes using training and testing data. At a glance, the ANN model was the best one of the prediction models, with the lowest scatter index (<0.1—excellent performance) and the shortest bars for OBJ parameters in both training and testing. Moreover, Figures 15 and 16 affirm that the ANN model had the lowest oscillation range of residual error compared to other models. All the mentioned figures showed that the predicted and measured values of compressive strength were more closed for the ANN model, indicating the superior performance of the ANN model compared to other models.

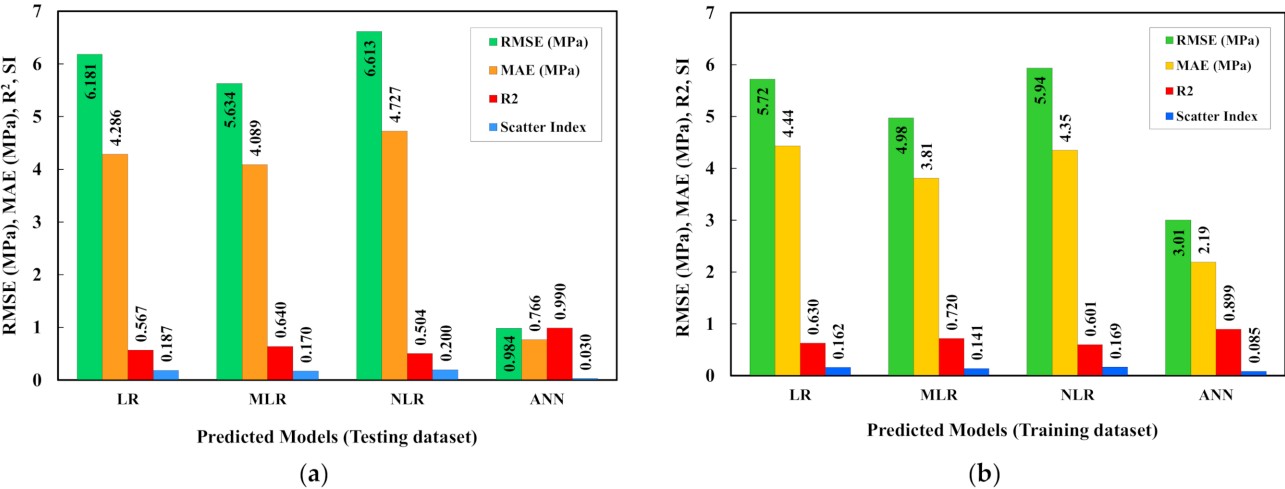

**Figure 13.** Statistical assessments to evaluate the efficiency of models used in this study. (**a**) Training dataset and (**b**) testing dataset.

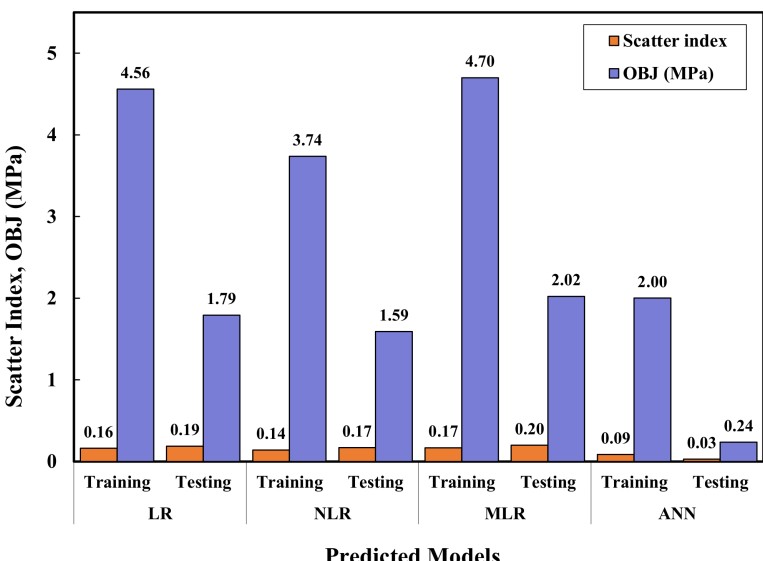

**Figure 14.** Superposition of measured and predicted value comparison between predicted models.

The OBJ value of training and testing for all proposed models is given in Figure 14 separately. The total OBJ (total of training and testing objective) values for LR, NLR, MLR, and ANN were 6.35, 5.33, 6.72, and 2.24, respectively, which also demonstrated overall that the ANN model was more efficient regarding the estimation of the compressive strength of paste mixtures modified with NS and MS.

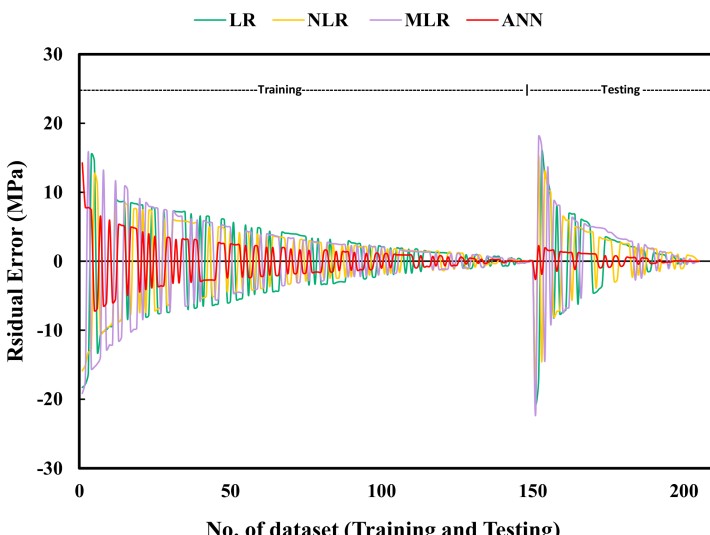

**Figure 15.** Residual error comparison of the models predicted the compressive strength of cement paste with nanosilica (NS) and microsilica (MS).

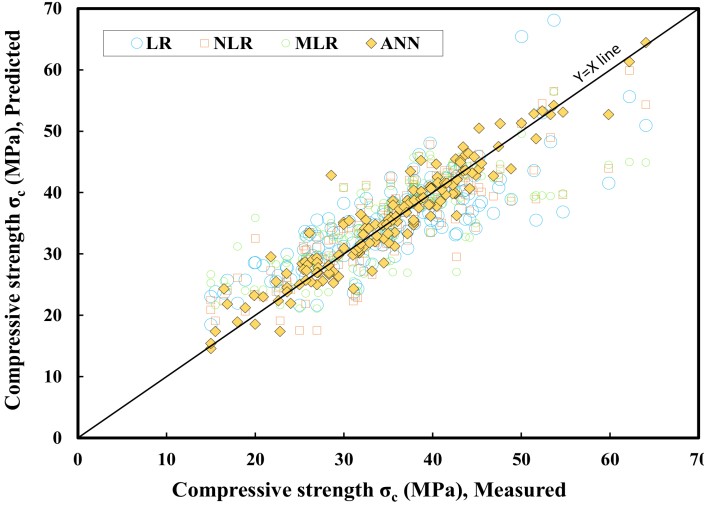

**Figure 16.** Superposition of measured and predicted value comparison between predicted models.

### 3.1.6. Sensitivity of Parameters

A sensitivity comparison for the models was conducted to find and evaluate the most impacting component affecting the compressive strength of paste mixtures adjusted with NS and MS [48]. By looking at the other equations, we could see that NS parameters were more significant than the MS additive, which shows NS's, effect to have been greater than MS. However, the most efficient model, ANN, was chosen for the sensitivity analysis. During the investigation, all the training datasets were used for each time, and a single input variable was extracted at a time for each set. For each attempt separately, the evaluation parameters such as R2, RMSE, and MAE were determined. Figure 10 summarizes the findings of the sensitivity analysis. The findings indicated that the curing time was the most critical and influential variable in predicting the compressive strength of paste mixes treated with NS and MS. This study also derived that the NS was more sensitive than the MS additive to the compressive strength of the cement-based paste. According to the obtained R2 value from that analysis, the NS sensitivity was 6.3% more than the MS additive.

It is worth mentioning that the further use of machine learning techniques in various application domains, e.g., foundation and pavements, can be of significant importance [50,51]. In this regard, using deep learning [52] could bring further novelty and complement our

research. However, one of the study's limitations was the life cycle assessment, reclaimed materials, and life cycle cost analysis for design innovation [52–55]. As mentioned, case studies on the specific application domains would be essential for the validation of the real-life performance as earlier investigated in the case of pavements [56–60]. Nevertheless, using novel machine learning methods such as hybrids and ensembles would be essential for future research for improving the model performance [61,62]. Training the machine learning methods with powerful evolutionary optimization techniques would be another approach to improving the models' quality [63–67].

## 4. Conclusions

Accurate and valid models for compressive strength prediction may result in substantial cost and time savings. According to the results achieved from this study, nanosilica highly affected the compressive strength of cement paste compared to microsilica. The curing time was the most significant parameter that caused an improvement in the compressive strength of the cement paste. The following findings could be drawn from the study and modeling of data from prior research used to predict the compressive strength of cement paste mixes modified with NS and MS in various combined proportions. Firstly, the relationship between the effective parameters on the compressive strength of cement paste mixtures was investigated with four different LR, NLR, MLR, and ANN models. Secondly, the ANN model worked better than other models with a greater $R^2$ value, lower RMSE, lower OBJ, and lower MAE values, and an excellent performance value of SI for training and testing datasets based upon the various evaluation parameters. In addition, the LR, NLR, MLR, and ANN models were the models built in this research to forecast the compressive strength of the cement paste. Based on the various evaluation parameters such as $R^2$, RMSE, MAE, SI, and OBJ, results indicated that the sequence of models was LR, NLR, MLR, and ANN, which meant that the ANN model was the best model proposed in this study based on data collected from literature, giving higher $R^2$ and lower RMSE and MAE values. SI values ranged from 0.03 to 0.2, suggesting good model results for all models and phases (training and testing). In addition, the ANN model had an SI value lower by 43.7% and 35.7% in the training process than the LR model and the NLR model. The OBJ value of the model for ANN was 46.5% lower than that of the NLR. This also shows that the ANN model was more effective for estimating the NS and MS compressive strength in paste mixtures. A sensitivity study showed that nanosilica was 6.3% more sensitive than MS as the most significant additive component for predicting paste mixtures' compression strength adjusted with nano or microsilica. In the future, research could be presented on the evaluation of the efficiency of different systematic models to predict the effect of the micro and nano size of $SiO_2$ on the porosity of cement mortar and paste as an essential factor on the compressive strength and durability of cementitious materials.

**Author Contributions:** Conceptualization, C.Y.R. and A.S.; methodology, C.Y.R. and A.S.; software, C.Y.R. and A.S.; validation, C.Y.R. and A.S.; formal analysis, C.Y.R. and A.S.; investigation, C.Y.R. and A.S.; resources, C.Y.R.; data curation, C.Y.R.; writing original draft preparation, C.Y.R.; writing review and editing, A.S. and A.A.B.; visualization, A.S.; supervision, A.S. and A.A.B.; project administration, A.S. and A.A.B. All authors have read and agreed to the published version of the manuscript.

**Funding:** This research received no external funding.

**Institutional Review Board Statement:** Not applicable for studies not involving humans or animals.

**Informed Consent Statement:** Informed consent was obtained from all subjects involved in the study.

**Conflicts of Interest:** The authors declare no conflict of interest.

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
