# Peer review of "Systematic Multiscale Models to Predict the Compressive Strength of Cement Paste as a Function of Microsilica and Nanosilica Contents, Water/Cement Ratio, and Curing Ages"

_sustainability, doi:10.3390/su14031723_

Round 1
Reviewer 1 Report
The article looks good to me
Author Response
Reviewer #1:
Comments and Suggestions for Authors
The article looks good to me
Response: Thank you very much for the observation made by reviewer #1.

Reviewer 2 Report
The purpose of the article entitled "Evaluation of the Efficiency of Different Systematic Models for Predicting the Effect of Micro and Nano Dimensions of SiO2 on the Compressive Strength of Cement Slurry" was know the effect of the various ratios of amorphous NS and MS on the compressive strength of cement paste as a function of water-cement ratio and curing time. Mathematical prediction models were used for training and testing to find out which one of NS and MS has a higher effect on the paste compressive strength. To apply the models, the authors use data from the literature.
- The article has writing issues. The text needs a complete review in the language. Also avoid using apostrophes (‘s).
- In certain excerpts, the author uses verbs in the gerund. The conjugated form must be used. For exemple:
“Several studies from previous experiments demonstrated this, decreasing the w/c substantially increasing the compressive strength of cement paste mixtures”,
Suggested: “Several previous experiments demonstrated this, in which decreases in the w/c ratio substantially increased the compressive strength of cement paste mixtures”;
- Write chemical compounds and other words that have subscripts correctly;
- In the introduction, verify: “along with the replacement of cement with supportive cement products (SCPs), namely Pozzolans”. Probably the correct one is: “Supplementary Cementitious Materials (SCMs)”. It is the most correct term for these materials added to cements.
- On line 182, the author is quoting MS, but I believe he changed it to the term NS. Review this part.
- Line 262) Unnumbered Section
- Lines 276-277) The structure of this sentence is not coherent
- As of line 297, there are some details on how the writing of the methodology should be carried out. I believe the author must have forgotten to delete the description, which seems to be the journal's standard. Lines 297-311) Paragraph not related to the study.
- Figure 3 is repeated.
- In Figure 14, in the last column, the author cites the acronym “MLP” as a model. Wouldn't it be ANN? (even on line 406 he talks about ANN, but on the chart is MLP). Review this part.
- Section 2.3) I don't think it is valid to present regression models in Figures 2 to 5. These models were built analyzing only one factor individually and ignoring the others. Since modeling considering all parameters has already been done in the next section I suggest removing the regression from the graphs and presenting only the data. A correlation matrix may be more useful in this section.
- ii) Curing Time) In Table 1 of the Appendix there are only 3 experimental data with 90 days curing time. The amount of data seems insufficient even for the proposed simple linear regression. Did this specific data enter the training set or the validation set?
- Section 3.1) The model in equation 2 is only one of the linear regression options. Since it is a comparative study wouldn't it be valid to explore interactions between the estimators and quadratic terms? Some of the variability in the linear regression can be attributed to interactions not evaluated.
- Equation 6) This is not the traditional formulation of the coefficient of determination.
- Section 5) Since the purpose of the paper is to compare different models, it is essential to present the summary of the ANOVA and significance of each estimator.
- Lines 321-323) I don't think it is necessary to describe operational procedure for determining regression coefficients. Citing the method (least squares) is enough here.
Author Response
Reviewer #2:
The purpose of the article entitled "Evaluation of the Efficiency of Different Systematic Models for Predicting the Effect of Micro and Nano Dimensions of SiO2 on the Compressive Strength of Cement Slurry" was know the effect of the various ratios of amorphous NS and MS on the compressive strength of cement paste as a function of water-cement ratio and curing time. Mathematical prediction models were used for training and testing to find out which one of NS and MS has a higher effect on the paste compressive strength. To apply the models, the authors use data from the literature.
- The article has writing issues. The text needs a complete review in the language. Also avoid using apostrophes (‘s).
Response: All apostrophes (‘s) were removed, and the text was reviewed completely. Most of the paraghraphs have been modified in terms of grammar. Thanks.
- In certain excerpts, the author uses verbs in the gerund. The conjugated form must be used. For exemple: “Several studies from previous experiments demonstrated this, decreasing the w/c substantially increasing the compressive strength of cement paste mixtures”,
Suggested: “Several previous experiments demonstrated this, in which decreases in the w/c ratio substantially increased the compressive strength of cement paste mixtures”;
Response: Thanks for the suggestion and the comments. The paragraph has been modified.
- Write chemical compounds and other words that have subscripts correctly;
Response: fixed, and thanks.
- In the introduction, verify: “along with the replacement of cement with supportive cement products (SCPs), namely Pozzolans”. Probably the correct one is: “Supplementary Cementitious Materials (SCMs)”. It is the most correct term for these materials added to cements.
Response: Done, and thanks.
- On line 182, the author is quoting MS, but I believe he changed it to the term NS. Review this part.
Response: That was a typo mistake, fixed, and thanks.
- Line 262) Unnumbered Section
Response: Corrected, and thanks.
- Lines 276-277) The structure of this sentence is not coherent
Response: Done and thanks.
- As of line 297, there are some details on how the writing of the methodology should be carried out. I believe the author must have forgotten to delete the description, which seems to be the journal's standard. Lines 297-311) Paragraph not related to the study.
Response: Removed and thanks.
- Figure 3 is repeated.
Response: Removed and thanks.
- In Figure 14, in the last column, the author cites the acronym “MLP” as a model. Wouldn't it be ANN? (even on line 406 he talks about ANN, but on the chart is MLP). Review this part.
Response: Agree with a comment, changed, and thanks.
- Section 2.3) I don't think it is valid to present regression models in Figures 2 to 5. These models were built analyzing only one factor individually and ignoring the others. Since modeling considering all parameters has already been done in the next section I suggest removing the regression from the graphs and presenting only the data. A correlation matrix may be more useful in this section.
Response: Done, and thanks.
- ii) Curing Time) In Table 1 of the Appendix there are only 3 experimental data with 90 days curing time. The amount of data seems insufficient even for the proposed simple linear regression. Did this specific data enter the training set or the validation set?
Response: Yes, the data has been divided into two parts training and testing, and the value of curing 90 days was included in both parts.
- Section 3.1) The model in equation 2 is only one of the linear regression options. Since it is a comparative study wouldn't it be valid to explore interactions between the estimators and quadratic terms? Some of the variability in the linear regression can be attributed to interactions not evaluated.
Response: Thanks for the comment and observation made by reviewer #2,
The linear regression model is used in this section because, “one dependent variable to one independent variable” which compressive strength only depends on w/c, which has been approved in Fig.2a that relationship is not accurate.
The multiple regression situation, b1, for example, is the change in Y relative to a one-unit change in X1, holding all other independent variables constant (i.e., when the remaining independent variables are held at the same value or are fixed). The word “ Multilinear could be used instead of the Nonlinear Regression Model since the compressive strength of the cement mortar depends on w/c ratio, curing time, and micro and nano silica contents.
Linear regression is a linear approach to modeling the relationship between a scalar response (or dependent variable) and one or more explanatory variables (or independent variables), and the researcher widely knows it.
The effect of the model parameters have been explained in the text with each model section, for example, if we use this model ( to predict the compressive strength of the cement paste from the parameter =-22.17 can be concluded that the water to cement ratio is the most influential parameter on decreasing the compressive strength of the cement paste. Then the second most influential parameter on increasing the compressive strength is NS content. So, from the model parameters will be able to know which cement mortar compositions affect the compressive strength of the cement paste.
The main conclusion of this study is no direct correlation was found between the compressive strength of the cement paste and the variables such as w/c, curing time, and micro and nano silica contents. So, the compressive strength of the cement paste depends on all the variables. A direct relationship has to be observed between compressive strength and the composition of cement paste, such as micro and nano-silica contents and w/c and curing time.
- Equation 6) This is not the traditional formulation of the coefficient of determination.
Response: Done, and thanks.
- Section 5) Since the purpose of the paper is to compare different models, it is essential to present the summary of the ANOVA and significance of each estimator.
Response:
Agree with a comment that the Analysis of variance (ANOVA) is a collection of statistical models, and their associated estimation procedures (such as the "variation" among and between groups) used to analyze the differences among means. And it is a useful analytical program, but it was not considered in this study. In the future ongoing study, the ANOVA will be used differently.
- Lines 321-323) I don't think it is necessary to describe operational procedure for determining regression coefficients. Citing the method (least squares) is enough here.
Response: Done, and thanks.

Reviewer 3 Report
Reviewed manuscript titled "Evaluation of the Efficiency of Different Systematic Models to Predict the Effect of Micro and Nano 2 Size of SiO2 on the Compressive Strength of Cement Paste" is very important from point of view of cementitious materials with additives of by products from another branches of industry. This paper has potential to be published after addressing the following comments:
- The title of the work should be revised since major theme of this work is associated with comparing the strength of NS and MS predicted with different models using the data obtained from literature.
- Workability of a matrix is compromised when MS and NS is used in a cement mix. to maintain the workability without compromising strength superplasticizers are added. I suggest the methodology should have also included this parameter as a variable in modeling.
- The cited literature should be improved with more current works, such as:
https://www.scientific.net/KEM.744.8
https://doi.org/10.3390/ma12142291
https://doi.org/10.1007/s11356-021-13918-2
- In Table 2, some parameters are written in bold letters, what does this indicate?
- English needs to be thoroughly reviewed.
Author Response
Reviewer #3:
Reviewed manuscript titled "Evaluation of the Efficiency of Different Systematic Models to Predict the Effect of Micro and Nano 2 Size of SiO2 on the Compressive Strength of Cement Paste" is very important from point of view of cementitious materials with additives of by products from another branches of industry. This paper has potential to be published after addressing the following comments: The title of the work should be revised since major theme of this work is associated with comparing the strength of NS and MS predicted with different models using the data obtained from literature.
Response: The title of the study has been modified based on your suggestion. Thanks
Workability of a matrix is compromised when MS and NS is used in a cement mix. to maintain the workability without compromising strength, superplasticizers are added. I suggest the methodology should have also included this parameter as a variable in modeling.
Response: I totally agree with the use of the superplasticizer contents in the model IF it has been used with cement paste mixes. But in this study, the dataset collected from the literature did not use a superplastizer in their studies.
The cited literature should be improved with more current works, such as:
https://www.scientific.net/KEM.744.8
https://doi.org/10.3390/ma12142291
https://doi.org/10.1007/s11356-021-13918-2
Response: thanks for recommending useful papers which have been cited in this paper.
In Table 2, some parameters are written in bold letters, what does this indicate?
Response: It should be noted that through the 28 trials conducted in WEKA with various numbers of hidden layers and neurons (Table. 2), the case with two hidden layers and 21 neurons (14, 7) was the best one with the highest R2 value and the lowest amount of RMSE and MAE compared to another, but due to equation complexity, the case with one hidden layer and 14 neurons was chosen, which had the best conditions through the one-layer trials. These two rows of parameters have been written in bold letters as the best trials.
English needs to be thoroughly reviewed.
Response: Whole paper including, The texts, abstract, tables, and figures, have been edited and organized in conversions, spelling, nouns, variety, word order, punctuations, prepositions, and fluency using editing software. The paper has been modified to satisfy the scientific journal requirements.

Reviewer 4 Report
This paper has many major flaws that must be corrected before further evaluation:
- Page 2, lines 64-66: The sentence “NPs have high pozzolanic characteristics due to finer particle volume” is too generic. Do all nanoparticles exhibit pozzolanic properties? Actually, many distinct strengthening mechanisms are associated with the different nanomaterials mentioned in the Introduction section (e.g., nano-core effect, bridging effect, pore refinement, nucleation effect, improvements of pozzolanic and reactions, etc).
- Page 2, line 75: The sentence “Micro/Nanosilica was found to accelerate the hydration process at early ages because of its considerably fine particle size” was not supported with reference to previous literature.
- Page 2, line 82: Why “setting time” was classified as “new characteristics”?
- In the introduction section, the authors could state the dimensions considered to classify SiO2 nanoparticles as microsilica (MS) or nano-silica (NS). Specific criteria for size classification should also be added to Section 2 (Table 1)
- Section 2.1 should be moved to the end of Section 1, as research significance is usually presented in the end of the “Introduction” section, rather than in the “Materials and Methods” section.
- All previous works dealing with MS or NS were listed in Table 1? If not, the authors must mention the methodology used to select the works cited in Table 1 (e.g., review strategies, keywords, databases, etc). The authors did not mention the criteria (e.g., journal, publication year, number of citations, main topics, etc) used in order to define whether a material is relevant to be analyzed.
- Table 1, Figures 2, 3, 4, and 5, all equations: contents of NS and MS were provided in terms of mass (or volume?) of cement? mass (or volume?) of composite? mass (or volume?) of binders? This information must be properly addressed, considering the methodology of each different work presented in Table 1.
- Table 1: water/cement ratio values were provided in terms of mass? Or volume?
- Page 3, line 130: how did the studies of Table 1 measure the “actual compressive strength”? Test methods, shape and size of specimens are expected to affect compressive strength results, which was not considered in this paper. All studies presented in Table 1 used similar standard methods to determine compressive strength?
- Section 2.3, items (i), (ii), (iii), (iv), and (v): the authors should not repeat and repeat the information presented in Table 1 and Figures 2, 3, 4, and 5. There should be some comparisons and synthesis of knowledge to provide new information, based on a critical overview of the statistical analyses of the collected data.
- Figures 2, 3, 4, and 5: The authors presented regression models with very low R² values (e.g., 0.2773, 0.3697, 0.215). Actually, they will not be useful, since the variability in the dataset is significantly large, so that those regression models were not a good fit for the experimental data. All of the elaborated models do a poor job of explaining and predicting the response variables. Then, there is not a reasonable justification to keep these models.
- Meaning of parameters a and b of Equation (1) was not mentioned.
- The statistical method used to determine the model parameters of Equations (1) and (2) was not mentioned.
- Page 9, line 262: Is “Criteria for evaluation of models” a subtitle?
- Meaning of many parameters of Equations 5-10 is not clear in the text (e.g., “Node1”, “Node2”, “Treshold”, “p”, “nall”, etc).
- The sentence “But for the R2 value, all other evaluation parameters have a best value of zero; but, R2 276 has the best value” should be rewritten for better comprehension.
- Page 9, lines 277-279: no idea why the authors used parentheses. Please check punctuation mistakes in the manuscript.
- -Section 4: the sentence “we can see that NS's parameters are more significant than the MS additive” refers to which parameters? How about the sentence “The findings indicate that curing time is the most critical and influential variable in predicting the compressive strength of paste mixes”? The reader may not understand these sentences in this location. The authors are discussing results before Section 5 (Results and Discussion), which does not make sense.
- Figure 16 was cited before Figures 6, 7, 8…, 15… It
- Page 10, lines 297-311: The authors presented 3 paragraphs of the “Microsoft Word template” of the journal as their own manuscript, which clearly demonstrates carelessness in the submission process.
- Table 2: meaning of R was not mentioned.
- Table 2: comma was used as decimal separator, which is not correct.
- Figure 10: Fonts of small sizes may be illegible for the readers.
- Page 18, line 454: The sentence “The compressive strength of cement paste mix models produced in this study was LR, NLR, MLR, and ANN” does not make sense.
- Conclusions: study limitations were not acknowledged.
- Conclusions: recommendations for further research on the topic were not provided.
- Consistency is another issue in this paper. For example, sometimes the authors have written “nanosilica” and “microsilica”, sometimes “nano-silica” and “micro-silica”, sometimes “Microsilics”…
- Poor English vocabulary was used in this paper. Vocabulary must be improved with specialized terminology associated with this scientific field (e.g., “composition materials” can be replaced by “composite materials”; “supportive cement products” can be replaced by “supplementary cementing materials”; “mix of cement” can be replaced by “cementitious matrix”; “cement structures” can be replaced by “concrete structures”, among many others).
- The authors did not provide the meaning of some abbreviations after their first appearance in the text of the manuscript (e.g., MS, NS)
- The authors did not use subscripts in chemical formulae to indicate the number of atoms of elements in different compounds (e.g., “CO2”, “Al2O3”, “Fe2O3 “).
- There are many typo mistakes (e.g., “yp”, “tp”, “tr”, “tst”, etc)
- There are many grammar mistakes (e.g., “there is not much evidence on which one is most effective”; “in two situations of training and testing to find out which one of NS and MS”; “which is why using”; “different modelization to (…) (ii) using statistical evaluation parameters, test and find (…)”; among many others).
Author Response
Reviewer #4:
Thanks for your time and effort in reviewing our paper, and your valid comments have been considered to improve the quality of the paper.
- Page 2, lines 64-66: The sentence “NPs have high pozzolanic characteristics due to finer particle volume” is too generic. Do all nanoparticles exhibit pozzolanic properties? Actually, many distinct strengthening mechanisms are associated with the different nanomaterials mentioned in the Introduction section (e.g., nano-core effect, bridging effect, pore refinement, nucleation effect, improvements of pozzolanic and reactions, etc).
Response: Agree with a comment. Some NPs have these characteristics. Pozzolans are a broad class of siliceous and aluminous materials that, when finely divided and in the presence of water, react chemically with calcium hydroxide (Ca(OH)2) at ordinary temperature to form compounds with cementitious properties. [ Mehta, P.K. (1987). "Natural pozzolans: Supplementary cementing materials in concrete". CANMET Special Publication. 86: 1–33.]
- Page 2, line 75: The sentence “Micro/Nanosilica was found to accelerate the hydration process at early ages because of its considerably fine particle size” was not supported with reference to previous literature
Response: The effect of Microsilica and nano-silica on the cement paste, which has been used in this study, are supported by this paper. [Shaikh, F., S. Supit, and P. Sarker, A study on the effect of nano silica on compressive strength of high volume fly ash mortars and concretes. Materials & Design, 2014. 60: p. 433-442].
The reference has been cited in the paper. Thanks
- Page 2, line 82: Why “setting time” was classified as “new characteristics”?
Response: It’s a mistake in typing, and the sentence has been modified. Thanks.
- In the introduction section, the authors could state the dimensions considered to classify SiO2 nanoparticles as micro silica (MS) or nano silica (NS). Specific criteria for size classification should also be added to Section 2 (Table 1) they are added as requested
Response: The range size of NS had been 30-100 nm, and about MS had been 0.2 μm and 2.32 g/cm3 particle density. The information was added in Table 1 as well.
- Section 2.1 should be moved to the end of Section 1, as research significance is usually presented in the end of the “Introduction” section, rather than in the “Materials and Methods” section.
Response: It’s replaced based on your suggestion.
- All previous works dealing with MS or NS were listed in Table 1? If not, the authors must mention the methodology used to select the works cited in Table 1 (e.g., review strategies, keywords, databases, etc). The authors did not mention the criteria (e.g., journal, publication year, number of citations, main topics, etc) used in order to define whether a material is relevant to be analyzed.
Response: The criteria for collecting data were added in the text as a track change.
(From 2017 to 2020, Cement mortar, Compressive strength test, experimental data, published in credible journals, containing MS and NS without any other additional material, same curing condition and with samples that they have been at least 3 or more different curing times.)
- Table 1, Figures 2, 3, 4, and 5, all equations: contents of NS and MS were provided in terms of mass (or volume?) of cement? mass (or volume?) of composite? mass (or volume?) of binders? This information must be properly addressed, considering the methodology of each different work presented in Table 1.
Response: Thanks for the comment, cement mortar with MS% + NS% substitution of mass of cement. The information was added in the methodology section as well.
- Table 1: water/cement ratio values were provided in terms of mass? Or volume?
Response: Water/cement ratios were mixed in terms of mass.
- Page 3, line 130: how did the studies of Table 1 measure the “actual compressive strength”? Test methods, shape and size of specimens are expected to affect compressive strength results, which was not considered in this paper. All studies presented in Table 1 used similar standard methods to determine compressive strength?
Response: Mortar cubes were determined according to IS 2250-1981, and their size in four papers [29,30,32,33] was 50*50*50 mm, and just in one of them [31] the size of the model (40*40*40) mm was used.
- Section 2.3, items (i), (ii), (iii), (iv), and (v): the authors should not repeat and repeat the information presented in Table 1 and Figures 2, 3, 4, and 5. There should be some comparisons and synthesis of knowledge to provide new information, based on a critical overview of the statistical analyses of the collected data.
Response: The reviewer's comment responded in the text.
- Figures 2, 3, 4, and 5: The authors presented regression models with very low R² values (e.g., 0.2773, 0.3697, and 0.215). Actually, they will not be useful, since the variability in the dataset is significantly large, so that those regression models were not a good fit for the experimental data. All of the elaborated models do a poor job of explaining and predicting the response variables. Then, there is not a reasonable justification to keep these models.
Response: The regression model has been removed from figures based on reviewer comments.
- Meaning of parameters a and b of Equation (1) was not mentioned.
Response: The model parameters in Eq. 1 have been mentioned in the paper. Thanks
- The statistical method used to determine the model parameters of Equations (1) and (2) was not mentioned.
Response: The mathematical equation parameters have been determined using solver functions in Excel by iteration, which give you a higher value of R2 and a low value of RMSE and MAE.
- Page 9, line 262: Is “Criteria for evaluation of models” a subtitle?
Response: This is a title for its following text, which explains the statistical criteria to evaluate the models. It’s corrected by numbering and bolding.
- Meaning of many parameters of Equations 5-10 is not clear in the text (e.g., “Node1”, “Node2”, “Treshold”, “p”, “nall”, etc).
Response: They are explained.
- The sentence “But for the R2 value, all other evaluation parameters have a best value of zero; but, R2 276 has the best value” should be rewritten for better comprehension.
Response: This expression is rewritten.
- Page 9, lines 277-279: no idea why the authors used parentheses. Please check punctuation mistakes in the manuscript.
Response: This parenthesis is removed and corrected.
- -Section 4: the sentence “we can see that NS's parameters are more significant than the MS additive” refers to which parameters? How about the sentence “The findings indicate that curing time is the most critical and influential variable in predicting the compressive strength of paste mixes”? The reader may not understand these sentences in this location. The authors are discussing results before Section 5 (Results and Discussion), which does not make sense.
Response: That’s a typing mistake, so this section is replaced overall by new text.
- Figure 16 was cited before Figures 6, 7, 8…, 15… It
Response: That’s corrected with item 18.
- Page 10, lines 297-311: The authors presented 3 paragraphs of the “Microsoft Word template” of the journal as their own manuscript, which clearly demonstrates carelessness in the submission process.
Response: it is removed with the correction the item 18.
- Table 2: meaning of R was not mentioned.
Response: The term R was explained in Table 2 based on reviewer comments.
- Table 2: comma was used as decimal separator, which is not correct.
Response: it is replaced with “+” and it’s explained.
- Figure 10: Fonts of small sizes may be illegible for the readers.
Response: it has been resized.
- Page 18, line 454: The sentence “The compressive strength of cement paste mix models produced in this study was LR, NLR, MLR, and ANN” does not make sense.
Response: This sentence is rewritten.
- Conclusions: study limitations were not acknowledged.
Response: Your valid comment regarding the conclusions part has been taken into account, and the conclusions have been modified with much more information. Thanks
- Conclusions: recommendations for further research on the topic were not provided.
Response: A recommendation for future work is added to the paper.
- Consistency is another issue in this paper. For example, sometimes the authors have written “nanosilica” and “microsilica”, sometimes “nano-silica” and “micro-silica”, sometimes “Microsilics”…
Response: They have been changed to nanosilica and microsilica.
- Poor English vocabulary was used in this paper. Vocabulary must be improved with specialized terminology associated with this scientific field (e.g., “composition materials” can be replaced by “composite materials”; “supportive cement products” can be replaced by “supplementary cementing materials”; “mix of cement” can be replaced by “cementitious matrix”; “cement structures” can be replaced by “concrete structures”, among many others).
Response: Based on this comment, the texts, including the abstract, tables, and figures, have been edited and organized in conversions, spelling, nouns, variety, word order, punctuations, prepositions, and fluency. The paper has been modified to satisfy the scientific journal requirements. More scientific information and details were added to the paper to be satisfied with the reader.
- The authors did not provide the meaning of some abbreviations after their first appearance in the text of the manuscript (e.g., MS, NS)
Response: They have been defined in the abstract and the keywords section, so they are conducted first in the manuscripts.
- The authors did not use subscripts in chemical formulae to indicate the number of atoms of elements in different compounds (e.g., “CO2”, “Al2O3”, “Fe2O3 “).
Response: Done and thanks.
- There are many typo mistakes (e.g., “yp”, “tp”, “tr”, “tst”, etc)
Response: They’re corrected.
- There are many grammar mistakes (e.g., “there is not much evidence on which one is most effective”; “in two situations of training and testing to find out which one of NS and MS”; “which is why using”; “different modelization to (…) (ii) using statistical evaluation parameters, test and find (…)”; among many others).
Response: Whole paper including, The texts, abstract, tables, and figures, have been edited and organized in conversions, spelling, nouns, variety, word order, punctuations, prepositions, and fluency using editing software. The paper has been modified to satisfy the scientific journal requirements.

Round 2
Reviewer 2 Report
No additional comments
Author Response
Thank you so much for your kindly comments.
Reviewer 4 Report
The authors improved the quality of the manuscript, but failed to solve the following issues:
- In response to comment 6 of Reviewer #4, the authors mentioned some criteria used to select experimental data. However, most of the methods were not mentioned in the manuscript (e.g., review of works published from 2017 to 2020, dealing with cement mortars containing MS and NS without other admixtures, subjected to the same curing conditions, etc).
- Table 1 should state that the water/cement ratios were provided in terms of mass.
- The manuscript must indicate the shape and size of mortar specimens and standard test methods used to determine the compressive strength values of references of Table 1 [29, 30, 31, 32, 33].
- The difference between some parameters of Equation 10 is not clear in the text (e.g., ntr, ntst, nall). The meaning of the subscripts should be indicated.
- Consistency is still an issue in this paper. For example, sometimes the authors used the term “Figure”. Sometimes, they used the term “Fig.”. Sometimes the authors wrote “microsilica”, sometimes “Microsilics”.
- The authors should be consistent in the use of capital letters in the title of the paper.
- Conclusions: limitations of the present paper were not acknowledged.
- Most of the sentences of the manuscript were not presented in an understandable way. English is very poor. The authors did not get someone expert in English to correct mistakes of the paper. This reviewer cannot recommend the publication of a paper containing such amount of grammar, punctuation, and typographical errors. All sentences must be improved by a native speaker, in order to reach a good level of English. After this review, the readers will focus on the contents of the paper, not on grammatical errors. I will limit myself to cite only some mistakes of the manuscript:
- There are many grammar and typo mistakes, such as “the most value of R2”; “for collecting tha data”; “the most change will mean the highest sensitivity”; among others.
- Technical vocabulary is very poor: “strength of compression” must be replaced by “compressive strength”; “modelization approaches” must be replaced by “modelling approaches"; “composition materials” must be replaced by “composite materials”; “cement structures” must be replaced by “concrete structures”; “adjusted with” must be replaced by “containing” or “incorporating”; among others.
Author Response
Manuscript ID: sustainability-1426449
Type of manuscript: Article
Dear Editor:
Thanks to the editor and reviewer comments, which have improved the work with more supporting information. The paper has been changed based on the review comments. All the reviewer suggestions (point by point) have been considered, and the paper's alterations are marked by track change.
Reviewer #4:
Comments and Suggestions for Authors
- In response to comment 6 of Reviewer #4, the authors mentioned some criteria used to select experimental data. However, most of the methods were not mentioned in the manuscript.
Response: The required criteria were added in the methodology in the manuscript.
- Table 1 should state that the water/cement ratios were provided in terms of mass.
Response: Done, and thanks.
- The manuscript must indicate the shape and size of mortar specimens and standard test methods used to determine the compressive strength values of references of Table 1 [29, 30, 31, 32, 33]c.
Response: Done, and thanks.
- The difference between some parameters of Equation 10 is not clear in the text (e.g., ntr, ntst, nall). The meaning of the subscripts should be indicated.
Response: Done, and thanks.
- Consistency is still an issue in this paper. For example, sometimes the authors used the term “Figure”. Sometimes, they used the term “Fig.”. Sometimes the authors wrote “microsilica”, sometimes “Microsilics”
Response: Done and thanks.
- The authors should be consistent in the use of capital letters in the title of the paper.
Response: Done and thanks.
- Conclusions: limitations of the present paper were not acknowledged.
Response: The whole paper, including the conclusions, has been revised, edited, restructured, and modified based on the reviewer's comment. The paper is satisfied the journal requirements after responding to the reviewer comments point by point.
Most of the sentences of the manuscript were not presented in an understandable way. English is very poor. The authors did not get someone expert in English to correct mistakes of the paper. This reviewer cannot recommend the publication of a paper containing such amount of grammar, punctuation, and typographical errors. All sentences must be improved by a native speaker, in order to reach a good level of English. After this review, the readers will focus on the contents of the paper, not on grammatical errors. I will limit myself to cite only some mistakes of the manuscript:
There are many grammar and typo mistakes, such as “the most value of R2”; “for collecting tha data”; “the most change will mean the highest sensitivity”; among others.
Technical vocabulary is very poor: “strength of compression” must be replaced by “compressive strength”; “modelization approaches” must be replaced by “modelling approaches"; “composition materials” must be replaced by “composite materials”; “cement structures” must be replaced by “concrete structures”; “adjusted with” must be replaced by “containing” or “incorporating”; among others.
Response: The texts, including the abstract, tables, and figures, have been edited and organized in terms of conversions, spelling, nouns, variety, word order, punctuations, preposition, and fluency using special software Editing (Grammarly). The paper has been modified to satisfy the scientific journal requirements.

Round 3
Reviewer 4 Report
The authors failed again to solve the following problems indicated in the first and second review steps:
- Standard test methods used by the authors of the works of Table 1 were not mentioned. If different methods are used, the compressive strength will change, so that the application of correction factors could be required. There is no discussion on this regard.
- Consistency is still an issue in this paper. For instance, sometimes the authors used the term “Figure”. Sometimes they used the term “Fig.”. Sometimes the authors wrote “microsilica”, while in Fig. 5a the word “Microsilics” was used.
- The software Grammarly was not able to detect and correct many English issues. The authors must get someone expert in English to correct mistakes of the paper. This reviewer cannot recommend the publication of a paper containing such amount of grammar, punctuation, and typographical errors. All sentences must be improved by a native speaker, in order to reach a good level of English. After this review, the readers will focus on the contents of the paper, not on grammatical errors. Again, I will limit myself to cite only some mistakes of the manuscript:
- There are many grammar and typo mistakes, such as “the most value of R2”; “for collecting tha data”; “the most change will mean the highest sensitivity”; among others.
- Vocabulary is very poor: “modelization approaches” must be replaced by “modelling approaches"; “composition materials” must be replaced by “composite materials”; “cement structures” must be replaced by “concrete structures”; “adjusted with” must be replaced by “containing” or “incorporating”; among others.
Author Response
Dear Reviewer,Thank you for your valuable comments and advice. We have followed all your comments and applied the corrections.
The revised sections are highlighted for your kind consideration.
In the methods section the description of all ML and statistical methods had been extended.
We have added proper citations to the methods.
The flowchart of research is presented.
We avoided adding confusing and extra sections and sub-sections. We have renamed the sections and used the standard ones.
We have extended the state of the art.
We added a paragraph of discussions on the validation and limitations.
Equations had been correctly numbered.
Native and professional proofreading had been applied